# MouseBytes, an open-access high-throughput pipeline and database for rodent touchscreen-based cognitive assessment

Flavio H Beraldo[1,2,3†], Daniel Palmer[1,4†], Sara Memar[1†], David I Wasserman[1,4†], Wai-Jane V Lee[1,2], Shuai Liang[5], Samantha D Creighton[4], Benjamin Kolisnyk[1,2‡], Matthew F Cowan[1], Justin Mels[1,2], Talal S Masood[1,2], Chris Fodor[1], Mohammed A Al-Onaizi[1,6§], Robert Bartha[1,7], Tom Gee[5], Lisa M Saksida[1,3,8], Timothy J Bussey[1,3,8], Stephen S Strother[5,9], Vania F Prado[1,2,3,6], Boyer D Winters[4*], Marco AM Prado[1,2,3,6*]

[1]Robarts Research Institute, The University of Western Ontario, Ontario, Canada; [2]Graduate Program in Neuroscience, The University of Western Ontario, Ontario, Canada; [3]Department of Physiology and Pharmacology, The University of Western Ontario, Ontario, Canada; [4]Department of Psychology and Neuroscience Program, University of Guelph, Guelph, Canada; [5]Rotman Research Institute, Baycrest Hospital, Toronto, Canada; [6]Department of Anatomy and Cell Biology, The University of Western Ontario, Ontario, Canada; [7]Department of Medical Biophysics, The University of Western Ontario, London, Canada; [8]Brain and Mind Institute, The University of Western Ontario, Ontario, Canada; [9]Department of Medical Biophysics, University of Toronto, Toronto, Canada

*For correspondence:
bwinters@uoguelph.ca (BDW);
mprado@robarts.ca (MAMP)

[†]These authors contributed equally to this work

Present address: [‡]Deep Genomics, Toronto, Canada; [§]Department of Anatomy, Health Sciences Center, Kuwait University, Kuwait City, Kuwait

**Abstract** Open Science has changed research by making data accessible and shareable, contributing to replicability to accelerate and disseminate knowledge. However, for rodent cognitive studies the availability of tools to share and disseminate data is scarce. Automated touchscreen-based tests enable systematic cognitive assessment with easily standardised outputs that can facilitate data dissemination. Here we present an integration of touchscreen cognitive testing with an open-access database public repository (mousebytes.ca), as well as a Web platform for knowledge dissemination (https://touchscreencognition.org). We complement these resources with the largest dataset of age-dependent high-level cognitive assessment of mouse models of Alzheimer's disease, expanding knowledge of affected cognitive domains from male and female mice of three strains. We envision that these new platforms will enhance sharing of protocols, data availability and transparency, allowing meta-analysis and reuse of mouse cognitive data to increase the replicability/reproducibility of datasets.

## Introduction

The public nature of research and increased rigor applied to research outputs have encouraged new approaches to enhance transparency, data sharing, and reproducibility (*Button et al., 2013*). Over the past 10 years, Open Science initiatives featuring increased data sharing and high-throughput automated data collection have increased the efficiency, quality, integrity and reproducibility of data gathering (*Johnson and O'Donnell, 2009*; *Rahman and Watabe, 2018*). In genomics, for example, researchers have made major progress in understanding the genetic basis of diseases by establishing

multi-research site consortia and by providing access to these data through different open repositories (*Boucas, 2018*; *Diehl and Boyle, 2016*; *Gerstein, 2012*). In neuroimaging, data sharing and large open-access databases have enabled the development of new analytic tools allowing researchers to address questions that could not be answered using single data sets (*Biswal et al., 2010*; *Poldrack and Gorgolewski, 2014*).

Recently, there have been several attempts to build databases of rodent behaviour data. The Jackson Laboratory has developed the Mouse Phenome Database, a repository of mouse data taken from several studies (*Grubb et al., 2009*). Additionally, resources such as the International Mouse Phenotyping Resource of Standardised Screens provide different pipelines for the characterisation of mouse lines (*Koscielny et al., 2014*). Although these databases represent a necessary and fundamental shift in the availability of data, these repositories provide only limited information on high-level cognitive testing in mouse models.

Conventional cognitive assessments in mouse models are subject to large variation (*Kafkafi et al., 2018*; *Wahlsten et al., 2003*), which may be in part the result of lack of automation. Additionally, the methodology used for cognitive assessments can significantly vary among research groups. For example, a recent analysis focusing on transgenic mouse models of Alzheimer's disease found a significant amount of variation in the parameters used in the Morris Water Maze, including pool size, pool temperature, number of trials per day, and number of acquisition days (*Egan et al., 2016*). Further evidence shows that even when protocol parameters are controlled for, different experimenters can still obtain different behavioural results in conventional behavioural tasks (*Chesler et al., 2002*; *Crabbe et al., 1999*; *Kafkafi et al., 2018*). Overall, there are converging domains of evidence to suggest that non-automated and non-standardised conventional behavioural assessments may be prone to several sources of bias.

Efforts to address these important gaps in data collection, automation, and translational research led to the development of the rodent touchscreen testing method (*Horner et al., 2013*; *Mar et al., 2013*; *Oomen et al., 2013*). This technology allows the use of tests in rodents that are highly similar, and in some cases identical, to human cognitive tests (*Heath et al., 2019*; *Nilsson et al., 2016*; *Nithianantharajah et al., 2015*; *Romberg et al., 2013a*). Touchscreen testing systems have standardised behavioural protocols that are under the control of a computer system, allowing for increased standardisation of outcomes. Furthermore, the automation of high-level cognitive testing can provide significant reductions in experimenter and environmental influence, by providing a standard operant environment and standardised output file formats (*Horner et al., 2013*; *Mar et al., 2013*). This feature makes results generated amenable to storage in a central repository, allowing for data categorisation, searching and comparison between multiple laboratories.

Here we used data obtained with male and female mice from three distinct mouse lines commonly used in Alzheimer's disease (AD) research to highlight the use of a new repository and Web-based software, MouseBytes (mousebytes.ca). We reveal longitudinal heterogeneity as well as commonalities in cognitive function between the various strains modelling AD. For example, 3xTG-AD mice, males and females, present early attention deficits (at 3–6 months of age) when compared to their age matched controls, demonstrating reproducibility of earlier results. Overall, our cognitive assessment suggests which mouse models can be used to model cognitive phenotypes consistent with Alzheimer's disease.

MouseBytes is available to researchers worldwide (mousebytes.ca), so they can pre-process, run automated quality control scripts, store, visualise, and analyse their data alone or alongside other researchers' stored data. Moreover, researchers can use a knowledge sharing tool https://touch-screencognition.org to disseminate community-driven information, including standard operating procedures (SOPs) and protocols. We foresee this repository for touchscreen data as a major step towards increasing the availability of datasets, including negative results, that can serve to evaluate reproducibility, decrease publication bias and to bring high-level cognitive assessment into the Open Science era.

## Results and discussion

### Open-access database and repository

To highlight the potential strengths of MouseBytes, we acquired data from male and female mice from three commonly used transgenic mouse models that have pathological similarities to AD at three ages in two different laboratories on three clinically-relevant touchscreen-based cognitive tests: attention [5-choice serial reaction time task (5-CSRTT)] (*Beraldo et al., 2015*; *Mar et al., 2013*; *Romberg et al., 2011*), behavioural flexibility [pairwise visual discrimination reversal (PD)] (*Bussey et al., 2008*; *Kolisnyk et al., 2013*; *Mar et al., 2013*) and long-term learning and memory [paired-associates learning (PAL)] (*Al-Onaizi et al., 2016*; *Bartko et al., 2011*; *Horner et al., 2013*). The mouse lines (3xTG-AD, 5xFAD and APP/PS1) were chosen due to their extensive use in AD research, as well as their differences in pathology development and AD familial genetic mutations (*Egan et al., 2016*; *Jankowsky et al., 2004*; *Lee and Han, 2013*; *Oakley et al., 2006*; *Oddo et al., 2003*). Moreover, the 3xTG-AD mouse line had been tested before using touchscreen attention tests providing a framework for reproducibility testing (*Romberg et al., 2011*).

Following completion of individual experiments, Extensive Markup Language (XML) files were generated using the Animal Behaviour Environment Test II (ABET II by Campden Instruments Ltd, Loughborough, England) software. XML files were uploaded into MouseBytes and screened using an automated quality control (QC) procedure which is a tool available at MouseBytes.ca (mousebytesQC). The rules and codes for the QC are available for download and modification in GitHub (GitHub_Touchscreen_Pipeline; copy archived at https://github.com/elifesciences-publications/Mousebytes-An-open-access-high-throughput-pipeline-and-database-for-rodent-touchscreen-based-data) (*Memar et al., 2019*) . Files that did not meet the QC criteria were automatically identified. Following QC, only XML files (one mouse unique ID/session) that passed QC were automatically uploaded to the database (mousebytes.ca) and integrated to the analytics TIBCO Spotfire to generate an interactive visualisation platform for 5-CSRTT, PAL, and PD (*Figure 1*, see also the online data visualisation - https://mousebytes.ca/data-visualization). Briefly, to navigate through the data in mousebytes visualisation (Spotfire) the users should select the cognitive task in the dropdown menu. After the selection of the cognitive task, corresponding features are selected (e.g. 5-SCRTT Probe trial for the 5-CSRTT data, etc.). A glossary with the description of the training and probe phases is found in MouseBytes (mousebytes.ca_description). Moreover, the user can check or uncheck the filter boxes on the right side of the page to define the data to be visualised and export or analyse specific graphs using the side tabs. This allows features selected, such as site, mouse strain, genotype, sex and age for example to be quickly compared (for more information on how to use the data visualisation please check the methods - Data Quality Control and availability).

Our pipeline enabled collection of extensive amounts of data from different ages, mouse strains, and sex. In total, we tested 652 different mice and generated 62,411 xml files (27,440 for 5-CSRTT, 17,230 for PAL, and 17,892 for PD). Importantly, by scanning the files through our automated QC procedure, we identified 487 files (0.8%) that did not meet the QC system criteria. 62,393 xml files (99.2%) passed the automated QC criteria and were transferred to the database. After QC, files that did not meet the criteria, or could not be fixed, were automatically discarded and were not used for analysis (see Materials and methods).

Due to the amount of data we generated in this work, classical graphical format to visualise all information contained in these datasets would require close to 30–40 figures with 6–10 panels each, depending on the kind of comparisons being performed. This situation underscores the need for online and on-the-fly data assessment, multidimensional visualisation, and comparison using online visualisation tools (*Dunn et al., 2016*), such as TIBCO Spotfire (*Dunn et al., 2016*; *Pechter et al., 2016*), which we used here.

Currently, our system has been optimised for the intake of data from the Bussey-Saksida Operant Chamber systems (Campden Instruments, Lafayette Instruments). There are alternative commercial touchscreen systems available (i.e. Med Associate K-Limbic touchscreen operant chambers), as well as several open source alternatives (*O'Leary et al., 2018*; *Pineño, 2014*; *Wolf et al., 2014*). In order to open the MouseBytes platform to all researchers using touchscreens, we have incorporated codes for download and modification in GitHub (GitHub codes; *Memar et al., 2019*) to easily convert the formats of output XML files from other systems to the format used in MouseBytes.

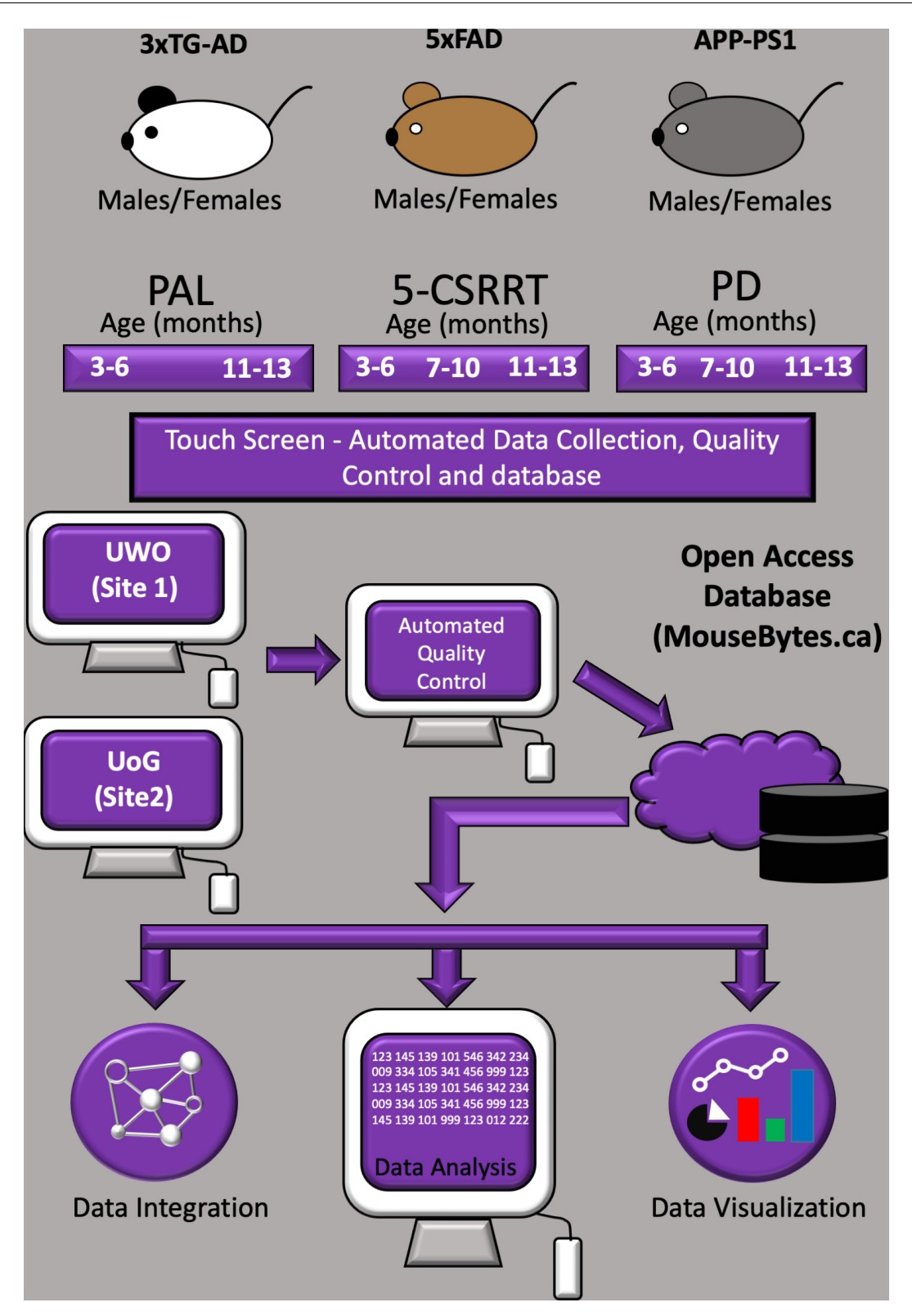

**Figure 1.** Schematic overview of the automated touchscreen cognition platform. Males and females of three different AD mouse lines were each evaluated in three different touchscreen tasks. The mice were food restricted and tested longitudinally and at two different sites (The University of Western Ontario (UWO) and University of Guelph (UoG) - Canada). Data were submitted to an automated QC process. Following automated QC, data were uploaded to an open-access database (mousebytes.ca) for post-processing analysis and visualisation using the analytics tool TIBCO Spotfire.
*Figure 1 continued on next page*

*Figure 1 continued*

The online version of this article includes the following figure supplement(s) for figure 1:

**Figure supplement 1.** Validation of image sets used during longitudinal testing.

## High-level cognitive testing in AD mouse models

The sample sizes for all experiments/tasks can be found in *Supplementary file 1*. Key parameters that were analysed for each experiment can be seen for 5-CSRTT: 5-CSRTT MouseBytes data link, PAL: PAL MouseBytes data link, and PD: PD mousebytes data link. One of the features of this open-access database is the possibility of downloading a standardised dataset (using a hyperlink generated by MouseBytes) related to particular experiments (i.e. linked to a particular figure of a paper) to perform customised analyses (A series of videos is available on the website that demonstrates how to use MouseBytes: MouseBytes-Guidelines).

Statistical analysis of the performance of distinct AD mouse models was conducted using R, taking advantage of the fact that CSV files can be generated for specific datasets using MouseBytes, which facilitates the use of open-source statistical packages. A summary of the split-plot ANOVA of all behavioural measures for each genotype can be found in *Supplementary file 2* (5-CSRTT), *Supplementary file 3* (PAL), and *Supplementary file 4* (PD). In addition, a second set of planned ANOVAs was conducted separately isolating cohorts by age and sex to identify potential genotype effects within select subpopulations of mice. Summary information with the complete secondary ANOVA statistics for all three tasks can be found in *Supplementary file 5*, *Supplementary file 6* and *Supplementary file 7*. A summary of the results of these statistical analyses can be found in *Table 1* (5-CSRTT) and *Table 2* (PD and PAL). Specific analyses and links to each dataset for figures are presented below.

## Reliability in touchscreen testing

Variability of mouse performance in behavioural tests across different laboratories is an important issue for replicability (*Crabbe et al., 1999*; *Kafkafi et al., 2018*; *Wahlsten et al., 2003*). The use of automated and standardized testing can help decrease variability, although a wide range of factors including colony genetic drift (*Zeldovich, 2017*), light-dark cycle, types of cages and housing (single housed or group-housed), source of food, type and amount of reward, different types of environmental enrichment and colony room temperature/humidity conditions can still potentially contribute to variability (*Kim et al., 2017*). Furthermore, even though touchscreen tasks are automated and standardised, there is some level of flexibility in these tasks. We are aware that researchers, depending on the scientific question, may modify the experimental design (set of images, length of inter-trial intervals, number of trials and sessions per day, type and or amount of reward, etc.), which can increase the number of variables for analysis. To control these variables, we included in MouseBytes

**Table 1.** 5-CSRTT analyses.

Summary of conventional genotype analyses on the 5-CSRTT task. Summary results were based on simple 2 (genotype) x 4 (stimulus duration) split-plot ANOVA. Impairment or Facilitation was determined by looking for a significant genotype effect or interaction. (3x – 3xTG-AD, 5x – 5xFAD and APP – APP/PS1) mouse lines. Impairment (↓), Improvement (↑) No Effect (-). See also *Supplementary file 2* and *5*.

| Sex | Age (months) | Accuracy 3x | Accuracy 5x | Accuracy APP | Omissions 3x | Omissions 5x | Omissions APP | Premature Responses 3x | Premature Responses 5x | Premature Responses APP | Perseverative Responses 3x | Perseverative Responses 5x | Perseverative Responses APP | Touch Latency 3x | Touch Latency 5x | Touch Latency APP | Reward Latency 3x | Reward Latency 5x | Reward Latency APP |
|---|---|---|---|---|---|---|---|---|---|---|---|---|---|---|---|---|---|---|---|
| Female | 3-6 | ↓ | - | ↑ | ↓ | - | ↑ | ↑ | - | - | ↑ | - | ↓ | ↓ | - | ↑ | - | ↓ | - |
| | 7-10 | ↓ | ↓ | - | - | - | ↑ | - | - | - | ↑ | ↑ | - | ↓ | - | - | - | ↓ | - |
| | 11-13 | ↓ | ↓ | - | - | - | - | - | - | - | ↑ | ↑ | - | ↓ | - | - | - | ↓ | - |
| Male | 3-6 | ↓ | - | - | - | - | - | - | - | - | - | - | ↓ | ↓ | - | - | ↓ | ↓ | - |
| | 7-10 | ↓ | - | - | - | - | - | - | - | - | - | ↑ | ↓ | ↓ | ↓ | - | ↓ | ↓ | - |
| | 11-13 | ↓ | ↓ | ↑ | ↓ | ↓ | - | - | - | - | - | ↑ | ↓ | ↓ | ↓ | - | - | ↓ | ↓ |

**Table 2.** PD and PAL analyses.

Summary of conventional genotype analyses on the PD and PAL tasks. Summary results were based on simple 2 (genotype) x 4 (stimulus duration) split-plot ANOVA. Impairment or Facilitation was determined by looking for a significant genotype effect or interaction. (3x – 3xTG-AD, 5x – 5xFAD and APP – APP/PS1) mouse lines. Impairment (↓), Improvement (↑) No Effect (-). See also **Supplementary file 3**, **4**, **6**, **7**.

| | | | Accuracy | | | Correction Trials | | | Touch Latency | | | Reward Latency | | |
|---|---|---|---|---|---|---|---|---|---|---|---|---|---|---|
| **Task** | **Sex** | | | | | | | | | | | | | |
| **Age (months)** | | | **3x** | **5x** | **APP** | **3x** | **5x** | **APP** | **3x** | **5x** | **APP** | **3x** | **5x** | **APP** |
| PD | Female | 3-6 | - | - | ↓ | - | ↓ | ↓ | ↓ | ↓ | ↑ | - | ↓ | ↑ |
| | | 7-10 | - | ↓ | - | - | ↓ | ↓ | ↓ | ↓ | ↑ | - | ↓ | ↑ |
| | | 11-13 | - | - | - | - | - | - | - | ↓ | ↑ | - | ↓ | - |
| | Male | 3-6 | ↑ | - | ↓ | ↑ | - | - | - | - | - | - | - | - |
| | | 7-10 | - | - | - | - | - | - | - | - | - | - | ↓ | - |
| | | 11-13 | - | - | - | - | - | - | ↑ | ↓ | - | - | ↓ | - |
| PAL | Female | 3-6 | - | - | ↑ | - | ↓ | ↑ | - | - | ↑ | ↑ | ↓ | ↑ |
| | | 11-13 | - | - | - | - | ↓ | - | - | ↓ | - | ↑ | - | ↓ |
| | Male | 3-6 | ↓ | - | ↓ | ↓ | ↓ | ↓ | ↓ | ↑ | ↑ | ↓ | ↓ | - |
| | | 11-13 | - | ↓ | - | - | ↓ | - | - | - | - | ↓ | ↓ | - |

features that allow the users to describe these variables as Metadata. For example, when uploading XML files, the user must check boxes indicating the light-cycle and whether mice were single or group housed. In addition, in experimental description users can describe how mice were tested (e.g. number of trials and sessions per day). Furthermore, users can also link the digital object identifier (DOI) of their published article to datasets. With these additional sources of metadata information, one can begin to determine which variables can influence behavioural performance within the touchscreen environment.

To directly assess potential site variability in the current dataset stored in MouseBytes, we included site as a factor in our 5-CSRTT analyses. The 5-CSRTT task was chosen due to the larger cohorts of mice used in these experiments at the two sites. Throughout the analyses of 5-CSRTT measurements (see Materials and methods), no consistent pattern of main effects or interactions emerged for site between age, sex, genotype, strain, or measure (stimulus length) (**Supplementary file 2**, tabs 1, 2 and 3). For example, for interactions between test site, genotype, sex, age and stimulus length, only APP/PS1 mice presented a statistical difference in accuracy, whereas all other parameters (% correct, number of premature responses, number of perseverative responses, reward collection latency and correct touch latency) for the three strains were not significantly different (**Supplementary file 2**, tabs 1, 2 and 3, lines 28 **#a**). Interactions with test site that were significant typically had a small effect size and lacked consistency across behavioural domains and mouse strains (**Supplementary file 2**, tabs 1, 2 and 3, **#a**). Overall, the evidence suggests low site-to-site variability and high replicability for touchscreen test performance. For example, low variability was observed between sites when we compared longitudinally the performance of wild-type female mice (B6129SF2/J) and their AD-mouse model counterpart, 3xTG-AD, in the 5-CSRTT task (attention). We observed a difference in accuracy (0.6 s stimulus duration) in wild-type females at 3–6 months of age between the two sites (**Figure 2A** dataset 1). Other than that, no statistically significant differences were found for either accuracy or omission for both B6129SF2/J females (**Figure 2B** dataset 2, **Figure 2C** dataset 3, **Figure 2D** dataset 4) or 3xTG females (**Figure 2E and F** dataset 5 and dataset 6, **Figure 2G and H** dataset 7 and dataset 8) between the two sites. We observed similar reproducible results for mouse touchscreen performance across the other AD models and their wild-type counterparts (see mousebytes.ca for more comparisons). As an important feature, MouseBytes allows the generation of dataset hyperlinks to easily identify and download the raw data used to generate each figure panel (dataset 1, 2, etc.).

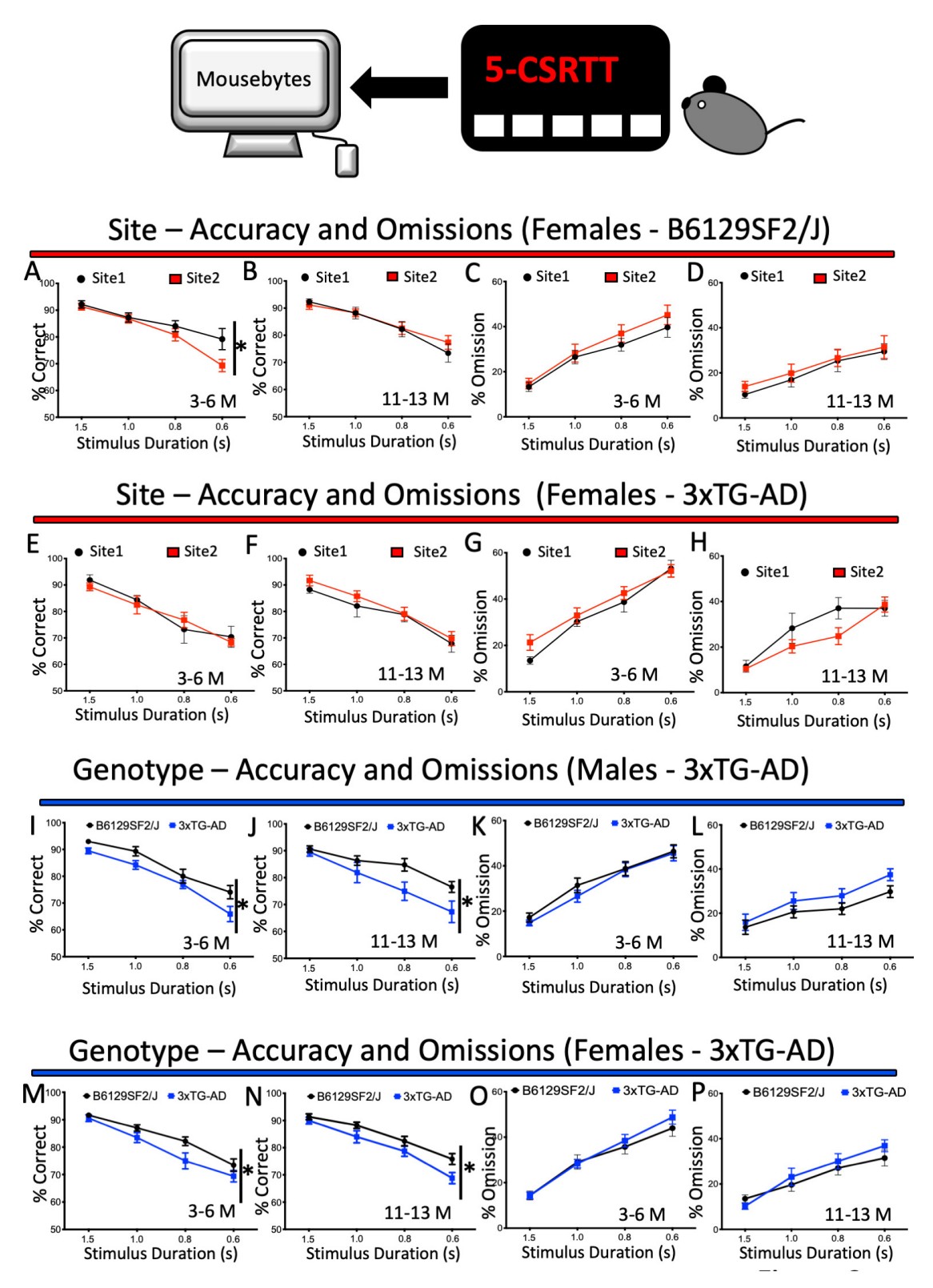

**Figure 2.** Performance and response measures of Male and Female mice during 5-CSRTT probe trials. Mice were subjected to a series of probe trials and the averages of accuracy (% correct), omissions (%) and premature responses (number) were plotted at different ages. The plots were generated with data downloaded from MouseBytes and the links (datasets) for the individual analysis can be found in the results section. (A-D), longitudinal site comparison of the performance (accuracy and omissions) of female Wild-type controls (B6129SF2/J) at 3–6 and 11 to 13 months of age; (E-H)

*Figure 2 continued on next page*

*Figure 2 continued*

longitudinal site comparison of the performance (accuracy and omissions) of female 3xTG-AD at 3–6 and 11 to 13 months of age respectively; (**I-L**) comparison of the performance (accuracy and omissions) of 3xTG-AD male and their Wild-type controls (B6129SF2/J) at 3–6 and 11 to 13 months of age; (**M-P**) comparison of the performance (accuracy and omissions) of 3xTG-AD female mice and Wild-type controls (B6129SF2/J) at 3–6 and 11 to 13 months of age. Results are presented as means ± s.e.m.; data were analysed and compared using Repeated measure Two-Way ANOVA and Bonferroni multiple comparisons post-hoc test; *p<0.05, compared with control.

The online version of this article includes the following figure supplement(s) for figure 2:

**Figure supplement 1.** PDEBrd1 mutation effect in the behaviour of 5xFAD mice.
**Figure supplement 2.** The effect of mild caloric restriction on Aβ(1–42) levels in male and female 3xTG and 5xFAD mice at 6 months of age.
**Figure supplement 3.** The effect of mild caloric restriction on amyloid pathology in male and female 5xFAD mice at 6 months of age.

Previous experiments have detected robust attentional deficits in 11- month-old male 3xTG-AD mice (*Romberg et al., 2011*), with lower accuracy in the 5-CSRTT and no differences in omissions compared to wild-type controls (*Romberg et al., 2011*). We tested male 3xTG-AD mice at the same age and reproduced the cognitive signature pattern of attentional deficit as previously published for male mice (*Figure 2J*, dataset nine for accuracy, *Figure 2L*, dataset 10 for omissions). In addition, we also tested female mice and similar to the males, 3xTG-AD female mice also presented lower accuracy (*Figure 2N* dataset 11) and no difference in omissions (*Figure 2P*, dataset 12) when compared to the wild-type controls. Moreover, both male and female 3xTG-AD mice that were tested starting at 4 months of age also presented lower accuracy (*Figure 2I*, dataset 13 and M, dataset 14) and no difference in omissions (*Figure 2K*, dataset 15 and 2 O dataset 16) when compared to controls (*Table 1* and *Supplementary file 2* and *Supplementary file 5* - Tab 1, *#b*). We also examined vigilance (the ability to maintain concentrated attention over a prolonged period of time), which was also previously reported to be affected in this mouse line (*Romberg et al., 2011*), by characterising performance across blocks of 10 trials. A complete breakdown of all the vigilance analyses can be found in *Supplementary file 8* (Tab 1). Reduced vigilance across trials was reflected in a deficit in accuracy in 3xTG-AD males (10–11 month-old mice used as example, *Figure 3A–D* and *Supplementary file 8*- Tab 1, *#c*) or 3xTG- AD female mice (3–6 -month-old mice used as example, *Figure 3I–L* and *Supplementary file 8* - Tab 1, *#c*). No differences were observed for omissions (*Figure 3E–H,M–P* and *Supplementary file 8* – Tab 1). These experiments support the replicability we observed between sites and suggest that 3xTG-AD mice present robust attentional deficits that can be observed across several laboratories even when a different genetic background is used. Because genetic drifting can potentially affect reproducibility in mouse behaviour testing (*Zeldovich, 2017*), identification of robust deficits of high-level cognition resulting from AD-related pathology is important to develop drug treatments. It seems that attention deficit in the 3xTG-AD is one such outcome.

The ability to compare mouse performance between sites can provide important insights on sources of variability for experiments. Replication experiments are the gold standard to validate scientific discoveries, but particularly in conventional rodent cognitive testing, variability of results is an issue. For example, different mouse models of AD present behavioural changes that are quite variable between laboratories when using conventional behaviour testing (*Arendash et al., 2001*; *Clinton et al., 2007*; *Ding et al., 2008*; *Holcomb et al., 1999*; *Ostapchenko et al., 2015*; *Stevens and Brown, 2015*). The combination of touchscreen cognitive testing and MouseBytes may help to identify sources of variability to overcome issues of replication in rodent high-level cognitive analysis.

In order to expand and further refine our understanding of the cognitive deficits in 3xTG-AD mice, we conducted two additional touchscreen-based cognitive assessments. Of particular interest was the PAL task, a relevant test for AD progression as the CANTAB version of PAL has been found to be predictive of conversion from mild cognitive impairment to AD (*Junkkila et al., 2012*). Moreover, forebrain cholinergic dysfunction, which is found in AD, impairs performance of mice in the PAL test (*Al-Onaizi et al., 2016*). We observed a small but significant deficit in the PAL task for 3xTG-AD male mice at four months of age (*Table 2*, *Supplementary file 3* and *Supplementary file 6* - Tab 1, *#d*). 3xTG-AD mice did not show any sign of deficits in visual discrimination learning or behavioural flexibility in PD (*Supplementary file 4* and *Supplementary file 7* - Tab 1). Overall, the cognitive phenotype of these mice resembled patients with early AD, presenting early deficits in

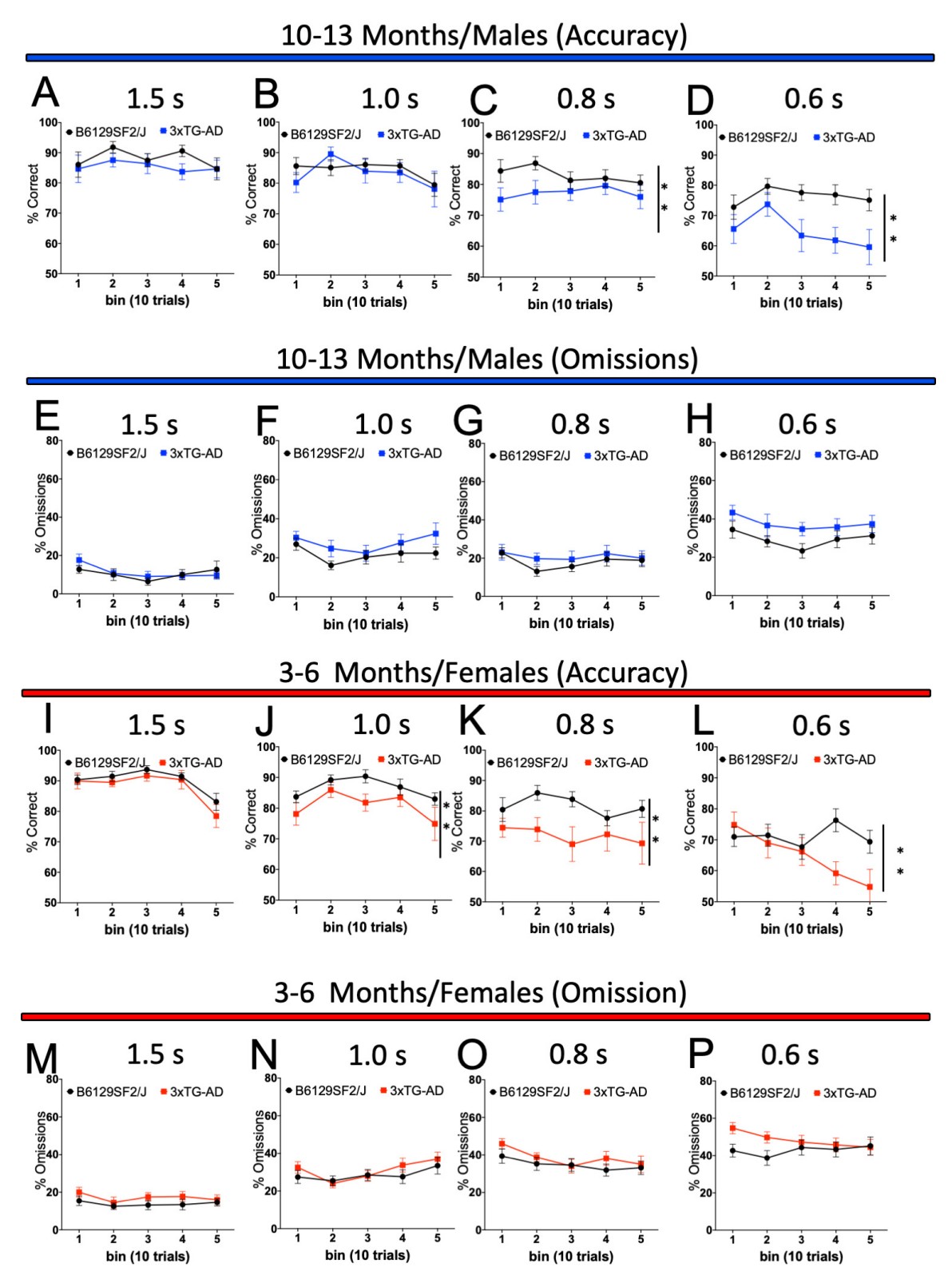

**Figure 3.** Sustained attention (vigilance) of 3xTG-AD male and female mice during the 5-CSRTT probe trials. Response accuracy and omissions in Wild-type and 3xTG-AD male (A-D for accuracy and E-H for omission) and female (I-L for accuracy and M-P for omission) mice were analysed across 10-trials blocks within the daily sessions of 50 trials with different stimulus durations. Results are presented as means ± s.e.m.; data were analysed and compared using Repeated measure Two-Way ANOVA and Bonferroni multiple comparisons post-hot test; **p<0.01 compared with control.

sustained attention (*Perry et al., 2000*) and visual-spatial learning (*Blackwell et al., 2004*), but not in behavioural flexibility (*Sahakian et al., 1989*).

In addition to testing 3xTG-AD mice, we also tested APP/PS1 and the widely used 5xFAD mouse line. We chose to use the 5xFAD mice in a mixed genetic background (C57Bl6 and Swiss Jim Lambert -SJL), as this was the original background in which this mouse line was generated (*Oakley et al., 2006*), and it is the most commonly used background across several laboratories (*Qosa and Kaddoumi, 2016*). However, the SJL genetic background presents the $Pdeb^{rd1}$ mutation that can lead to retinal degeneration (*Clapcote et al., 2005*) (see Materials and methods), which causes severe visual impairment in homozygosis. Given that some of the 5xFAD mice could be heterozygous for the $Pdeb^{rd1}$ mutation, we evaluated whether carrying one $Pdeb^{rd}$ allele affected the performance of mice in touchscreens using the PD task, which directly measures visual discrimination. The performance of mice carrying one $Pdeb^{rd}$ allele did not differ from those who did not (*Figure 2—figure supplement 1A, B and C*). Moreover, as touchscreen testing requires food restriction for motivation, we also assessed whether the food restriction protocols used for touchscreen cognitive testing affected amyloid production in 3xTG-AD and 5xFAD mice. Ultimately, we failed to find any differences in amyloid production and deposition between food restricted and non-food restricted animals (*Figure 2—figure supplement 2* and *Figure 2—figure supplement 3*).

The 5xFAD transgenic mouse line displayed a complex cognitive phenotype. Female 5xFAD mice displayed deficits in sustained attention that begin at 7 months in the 5-CSRTT task, while males show deficits by 11 months (*Table 1*, *Supplementary file 2* and *Supplementary file 5* - Tab 2, *#e*). Initial training on the PAL task did not generate robust results as both 5xFAD and their controls (both male and female) were poor performers (*Supplementary file 3* and *Supplementary file 6* – Tab 2). This highlights the utility of MouseBytes in assessing cognitive testing of a given mouse line by comparing with other lines in the database. However, a simplified version of the test revealed significant visual-spatial deficits at 10 months of age for both male and female 5xFAD mice (*Supplementary file 3* and *Supplementary file 6* – Tab 2, *#f*). No deficits in behavioural flexibility or visual discrimination learning were observed for 5xFAD mice when compared to their respective controls (*Supplementary file 4* and *Supplementary file 7* – Tab 2). The 5xFAD mouse line displays a subtler behavioural phenotype than the 3xTG-AD, but is still consistent with the impairments observed in AD. Interestingly, amyloidosis has been reported to start earlier and to be more aggressive in the 5XFAD (*Oakley et al., 2006*) compared to the 3xTG-AD line (*Oddo et al., 2003*). However, our results showed earlier deficits development in 3xTG-AD mice compared to the 5xFAD line, suggesting that this could be related to abnormal Tau function in the 3xTG-AD mouse line (*Oddo et al., 2003*).

APP/PS1 mice (male or female) did not show any sign of attentional deficits in the 5-CSRTT task at any age (*Supplementary file 2* and *Supplementary file 5* – Tab 3) similar to what was observed independently in a different background for this strain (*Shepherd et al., 2019*). However, APP/PS1 male mice presented with an early deficit in visual-spatial integration learning in the PAL task, which is consistent with the 3xTG-AD mouse phenotype (*Supplementary file 3* and *Supplementary file 6* – Tab 3, *#g*). Furthermore, an early deficit in behavioural flexibility was observed for female APP/PS1 mice at four months of age in the PD reversal task, which is interesting from the point of the translational utility of this mouse model, as behavioural flexibility deficits are not typically associated with AD at early stages of the disease progression (*Sahakian et al., 1989*) (*Supplementary file 4* and *Supplementary file 7* – Tab 3, *#h*).

Although each touchscreen task is generally run across labs using the same set of task-specific stimuli, task stimuli are being optimised continuously (*Horner et al., 2013*; *Mar et al., 2013* ). Furthermore, some researchers have run tasks multiple times within a cohort of mice using different stimuli (*Bartko et al., 2011*) and have found that the performance of animals may vary with the stimulus set used (*Bussey et al., 2008*). We extracted cross-site data from mice using different stimulus sets to show that indeed, depending on the image set used in PD or PAL, mice can reach higher or lower levels of discrimination accuracy (*Figure 1—figure supplement 1*). Our data indicate that, for PAL and PD or other touchscreen tasks using complex visual stimuli, longitudinal testing should be preceded by appropriate control experiments to avoid potential bias with image sets. We anticipate that when more data are available in MouseBytes, the touchscreen community will be able to compare a larger number of images sets and identify optimal stimulus combinations.

## Genetic background and touchscreen performance

The choice of background strain for mouse models of disease can have major implications for cognitive assessment (*Sittig et al., 2016*). However, due to the absence of framework within which to aggregate behavioural data, comparison of the performance by different mouse strains has been limited. For example, in previous work data acquisition needed to be standardised across laboratories to gather information on how genetic background influences performance (*Graybeal et al., 2014*). We compared the performance of mice from three different wild-type strains in the initial dataset deposited in MouseBytes (B6129SF2/J, B6SJLF1/J and C57BL6/j background). To directly assess strain variability in touchscreen test performance, data from 5-CSRTT experiments were extracted from MouseBytes and analysed (similar analyses can be performed for other tests by extracting the datasets from MouseBytes). Interestingly, both female (*Figure 4A*, dataset 17) and males (*Figure 4C*, dataset 18) B6129SF2/J presented higher levels of accuracy on 5-CSRTT at 3–6 months of age, but not at 11–13 months of age (*Figure 4B*, dataset 19 and 4D, dataset 20), when compared to the other two wild-type strains tested (B6SJLF1/J and C57BL6/j). Moreover, both male and female B6SJLF1/J mice (3–6 and 11–13 months of age) were found to engage in more premature responses than the B6129SF2/J and C57BL6/j lines (4E-H, datasets 21, 22, 23 and 24). This suggests a general phenotype of impulsiveness inherent to these B6SJLF1/J mice. We envision that with multiple users depositing their data in MouseBytes, it will be relatively easy to make comparisons of performance for thousands of mice from different strains. Ultimately, these overarching analyses could help to inform the background strains to be used for new mouse lines to investigate specific high-level cognitive domains, for example, models that can now be generated using new genome-editing techniques such as CRISPR/Cas.

## Sex variability

Recognition that behavioural rodent research is biased towards using male mice has led funding institutions to establish specific guidelines in the choice of animals for research (*McCullough et al., 2014*). Several neurobiological differences are present between male and female brains (*Grissom and Reyes, 2019*; *Ruigrok et al., 2014*). In the scope of AD, there are sex differences in the pathological development of plaques and tangles (*Corder et al., 2004*) and sex steroid hormones' levels can contribute to some of these effects (*Carroll et al., 2010*; *Carroll et al., 2007*). To highlight the potential for sex comparisons in high-level cognitive assessment, we initially compared 3xTG-AD mice, a mouse line that presents sex variability in pathology (*Carroll et al., 2010*; *Carroll et al., 2007*). We found no major differences in attentional performance (accuracy) when we compared male and female mice in 5-CSRTT, as shown for the B6129SF2/J at 3–6 months (*Figure 4I*, dataset 25) or 11–13 months of age (*Figure 4J*, dataset 26). Similarly, no differences were found when 3xTG-AD males and females were compared at 3–6 (*Figure 4K*, dataset 27) or 11–13 months of age (*Figure 4L*, dataset 28). These results suggest little or no difference for high-level cognitive performance between male and female mice. To the best of our knowledge, the dataset presented here and deposited in MouseBytes provides the most extensive evaluation of performance of female mice in touchscreen tests.

## Unbiased analysis of behavioural performance

While conventional ANOVA-based statistics can be employed to measure changes in behaviour, large datasets can make this approach difficult. In order to address these challenges, alternative types of analyses may be necessary. One potential solution is to employ machine learning algorithms or artificial intelligence systems. To provide an example of these high-level analyses to describe large behavioural datasets that can be extracted easily from MouseBytes, we generated a summary of the touchscreen behavioural data utilising a k-mean classification approach. This approach represents a class of unsupervised learning algorithms that can identify group clustering without any bias towards the behavioural measures or the sample identification such as of the genotype, age, sex, or test site. Recently, researchers have used longitudinal k-mean algorithms to subcategorise different cohorts of AD patient populations that had been previously classified in one large group based on disease progression (*Genolini et al., 2016*). We used the R package *kml3d* (*Genolini et al., 2013*) to conduct longitudinal k-mean unsupervised grouping of all the data for the three mouse lines. For all tasks, data were grouped into three categories loaded onto a similar progression of behavioural

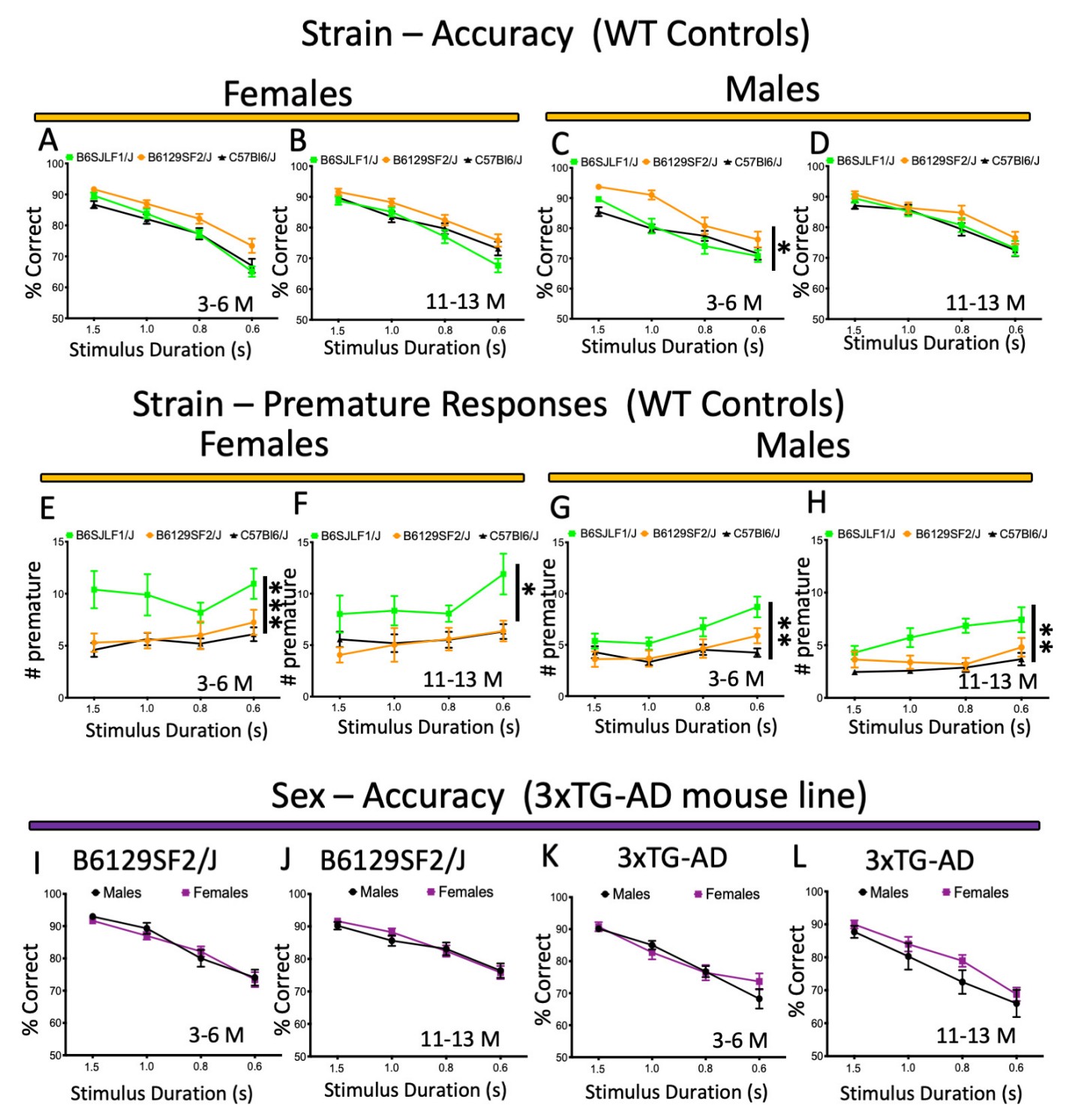

**Figure 4.** Performance and response measures of male and female mice during the 5-CSRTT probe trials. (**A-D**) Strain/mouse background comparison (accuracy) of female and male Wild-type controls (B6129SF2/J, B6SJLF1/J, C57Bl6/J) at 3–6 and 11–13 months of age; (**E-H**) Strain/mouse background comparison (premature responses) of female and male Wild-type controls (B6129SF2/J, B6SJLF1/J, C57Bl6/J) at 3–6 and 11–13 months of age; (**I-L**) Sex comparison (accuracy) of B6129SF2/J and 3xTG-AD females and males at 3–6 and 11–13 months of age. Results are presented as means ± s.e.m.; data were analysed and compared using Repeated measure Two-Way ANOVA and Bonferroni multiple comparisons post-hot test; *p<0.05, **p<0.01 and ***p<0.001 compared with control.

metrics of high performance, moderate performance, and low performance (*Figure 5—figure supplement 1*). We decided to choose three groups across all of our tasks in order to be consistent across domains, as well as to try and capture more subtlety in the clustering of behavioural characteristics. Because we tested animals at two or three temporally separated intervals during our cognitive testing, we decided to treat each animal's observations for each testing interval as independent. This was done to allow for mice to change membership across our K-mean groups. We expected that changes in k-mean group membership would likely indicate cognitive decline in our animal populations. Following k-mean grouping, we then used Fisher's exact test to determine if significant differences existed in the k-mean group membership between transgenic mice and their respective control strains. In order to separate out potential sex effects from genotype variation, we conducted comparisons separately between male and female mice for these analyses. Because animals were considered independent across age, we also conducted the analyses separately for each testing period. In order to account for multiple comparisons with Fisher's Exact Test, the Benjamini-Hochberg correction for false discovery rate was applied (*Table 3*). Visualisation of the k-mean group memberships by strain, task, age, and sex can be found in *Figure 5*.

In the PAL task, the behaviour of high performing mice was characterised by high accuracy (% correct) and low numbers of correction trials (*Figure 5—figure supplement 1A and B*). Mid-performing mice had reduced accuracy and slow correct and reward collection response latencies (*Figure 5— figure supplement 1A,C and D*). The low-performing mice showed consistently low accuracy and high numbers of correction trials (errors) (*Figure 5—figure supplement 1A and B*). The number of mice belonging to each cluster can be found in *Figure 5—figure supplement 1E*. No significant differences in k-mean group membership were found for the 3xTG-AD mice at any age (*Figure 5* and *Table 3*). Differences in membership were found to be significant for 5xFAD mice at 10–11 months of age, as more males 5xFAD transgenic were found to be in the low performing group, while more females 5xFAD transgenic were found to be in the mid performing group (*Table 3*). Interestingly, at four months of age, most 5xFAD wild-type and transgenic mice were low performers, suggesting the background strain may affect performance on this task, confirming our observation with traditional analysis. Significant group membership differences were observed for APP/PS1 mice at four months of age, and more female transgenic mice were found in the high performing group (*Figure 5*), while male APP/PS1 transgenic mice were in the low performing group (*Figure 5* and *Table 3*). These data suggest that while the PAL task might be a good behavioural predictor in human AD, further studies should be conducted to ensure that this effect is consistently observed across multiple AD mouse models. This is consistent with the small effect sizes in PAL, except for 10-month-old 5xFAD mice (*Supplementary file 3* and *Supplementary file 6*, Tab 2).

In the PD tasks, behaviour of high performing mice was characterized by high response accuracy (% correct) and low number of correction trials (errors) (*Figure 5—figure supplement 1F and G*).

**Table 3.** p-values from Fisher Exact Test for K-Mean Group.
Group Differences between wildtype and Transgenic mice across behavioural experiments. Fisher's exact test was conducted to compare the % membership of the high, mid, and low k-mean groups between wildtype and transgenic mice for each strain, sex, and age. All p-values have been adjusted with the Benjamini and Hochberg Correction.

| Task | Age | 3xTG-AD | | 5xFAD | | APP/PS1 | |
| | | Female | Male | Female | Male | Female | Male |
| --- | --- | --- | --- | --- | --- | --- | --- |
| 5-CSRTT | 3-6 | .03 | .01 | .03 | .06 | .17 | .13 |
| | 7-10 | .19 | .06 | .02 | .01 | .77 | .07 |
| | 11-13 | .01 | .39 | .02 | <0.001 | 1.00 | .19 |
| PD | 3-6 | .34 | .25 | <0.001 | .92 | .02 | .69 |
| | 7-10 | .81 | .16 | <0.001 | .06 | .42 | .61 |
| | 11-13 | 1.00 | .36 | <0.001 | .03 | .18 | .81 |
| PAL | 3-6 | .59 | .19 | .33 | .19 | .06 | .03 |
| | 11-13 | .77 | .95 | .03 | .01 | 1.00 | .19 |

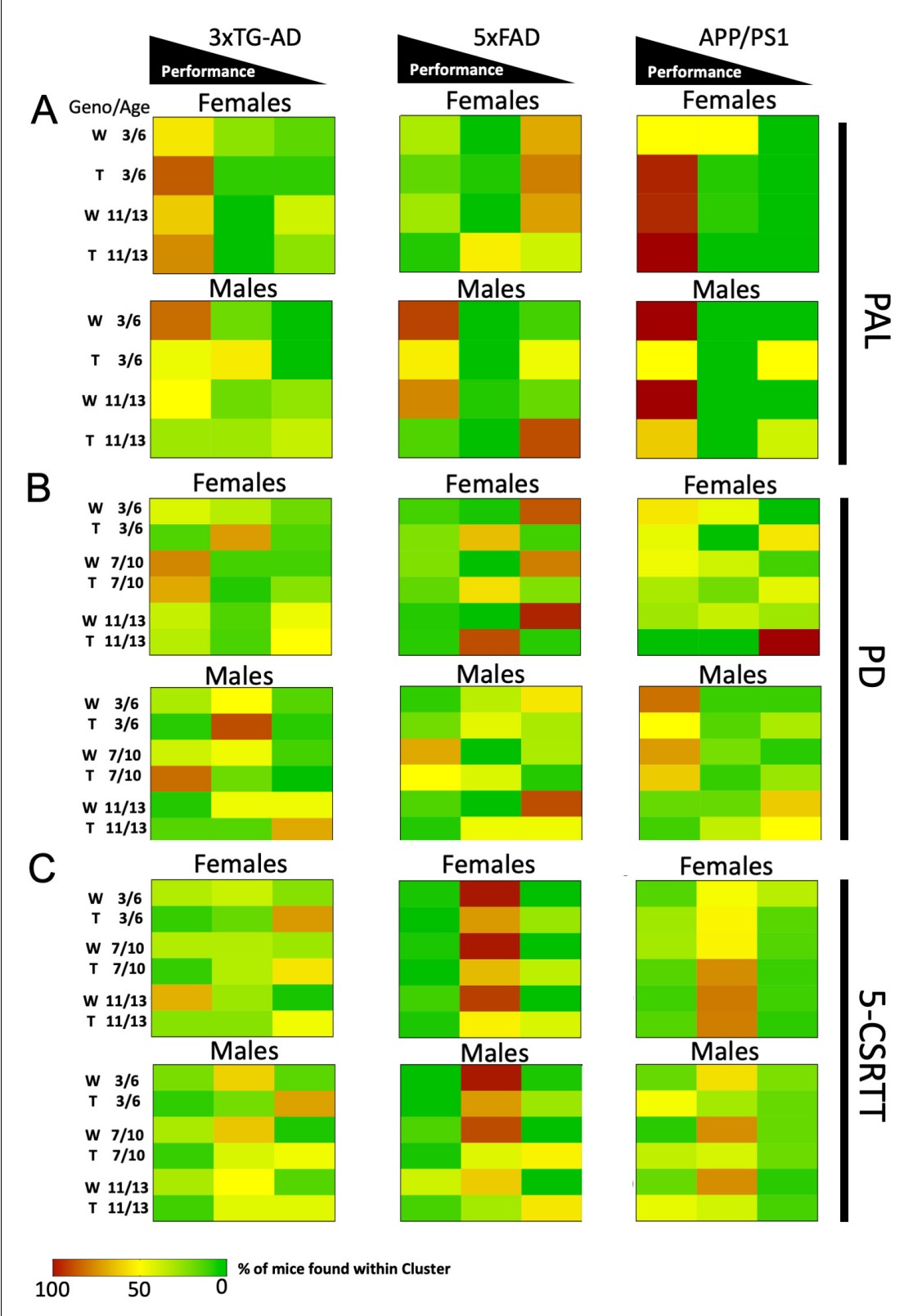

**Figure 5.** Heatmap visualisations of k-mean group membership across experiments. The percentage of group representation per strain is shown by the cell color. Cell color closer to red indicates a higher representation of mice in the k-mean grouping. Mice are divided by sex (male and female), genotype (W for wild type and T for transgenic), and age (3–6, 7–10, and 11–13). Analysis of the PAL (A) task data uncovered early group membership variability in the APP/PS1 mouse line, as well as later group membership differences in the male 5xFAD mice. In the PD (B) experiment, only APP/PS1

*Figure 5 continued on next page*

*Figure 5 continued*

transgenic mice showed an increase in membership of the low performing group compared to wildtype. See also S5. In the 5-CSRTT (**C**), transgenic 3xTG-AD and 5xFAD mice were found to shift to the lowest performing group of mice.

The online version of this article includes the following figure supplement(s) for figure 5:

**Figure supplement 1.** Results of longitudinal clustering analysis of PAL, PD and 5-CSRTT data. z-scores for parameters assessed during the 5-CSRTT, sorted by cluster.

The typical behaviour of the mid performing mice included slower correct and reward collection response latencies to the test stimuli (*Figure 5—figure supplement 1H and I*). Low performing mice showed a pattern of low accuracy (% correct) and high number of correction trials (*Figure 5—figure supplement 1F and G*). The number of mice belonging to each cluster can be found in *Figure 5—figure supplement 1J*. No significant group composition differences were observed for 3xTG-AD mice (*Table 3*). Fisher's exact test revealed significant differences in k-mean group composition for 5xFAD mice as more transgenic mice were found to occupy the mid performing group, while wild-type control mice occupied the low performing group (*Table 3*). Only a significant difference in group composition for APP/PS1 females was observed at four months, as there was a larger group of transgenic mice in the low performing group compared to wild type (*Table 3*). Overall, the different pattern of results in PD suggests it was not particularly sensitive to AD-related pathological changes. Separation of performance between strains was only observed for the female APP/PS1 mice.

In the 5-CSRTT, high performing mice were characterised by high accuracy (% correct), low omissions, and higher perseverative responses, *Figure 5—figure supplement 1K,L and M*). Mid performing mice had average response accuracy and rates of omissions (*Figure 5—figure supplement 1K, L*), but showed an increase in premature responses (*Figure 5—figure supplement 1N*). The low performing mice showed low accuracy, high omission, slow response times and slow reward collection latency (*Figure 5—figure supplement 1K, L, O and P*). The number of mice belonging to each cluster can be found in *Figure 5—figure supplement 1Q*. Significant k-mean membership differences were observed for 3xTG-AD mice, consistently at 4 months of age, and transgenic mice were usually clustered as low performers (*Figure 5* and *Table 3*). Fisher's exact test analysis revealed significant k-mean membership differences for 5xFAD transgenic mice and their respective control mice across all ages (*Table 3*), and the 5xFAD mice presented consistently low performance. These results suggest that both 5xFAD and 3xTG-AD transgenic mice consistently are lower performers (shifted to the lower performance group) than their WT counterparts (shifted to the higher and moderate performers) in the 5-CSRTT, suggesting that this test might be a good candidate for screening cognitive symptoms in these two mouse models of AD. Interestingly, these differences were not observed for the APP/PS1 mice. In fact, there was no difference in the performance of females APP/PS1 mice (3–6, 7–10 or 11–13). However, surprisingly, APP/PS1 male mice tended to shift their performance to the higher and moderate performers while the WT shifted to the lower performers.

Curiously, across all touchscreen paradigms, male and female 5xFAD mice had a consistent phenotype displaying increases in reward collection latency (*Tables 1* and *2*, *Supplementary file 2*, *3*, *4*, *5*, *6* and *7* – Tab 2, #*i*), suggesting the potential that these mice present lack of motivation or abnormal motor function at the ages tested, which has been described previously (*O'Leary et al., 2018*).

Collectively, the data show different patterns of cognition abnormalities between the three AD models, which may be related to different human AD mutations and the pathophysiology associated with them, including the tau mutation in the 3xTG-AD mouse line. Overall, two of the three lines showed a consistent deficit in attention and all lines presented modest but significantly lower performance in PAL. In addition, we also observed sex dissimilarities in PAL for 3xTG-AD and APP/PS1 (*Supplementary file 3*, tabs 1 and 3 #d and #g respectively), which could be due to differences in cellular and molecular mechanisms in brain development (*Grissom and Reyes, 2019*; *Ruigrok et al., 2014*) and/or the differences of AD-type pathology, disease onset and progression rate in males and females.

## Conclusions and next steps

Here we introduce an open-access high-throughput pipeline and a Web application database that facilitates data repository, searching, and analysis of touchscreen data. The MouseBytes data integration platform introduces quality control of high-throughput approaches using touchscreen analysis in an open source platform for dissemination of high-level cognitive data. Including standardised data from different laboratories around the world will bring the advantages of open-access data sharing and greatly enhance validation, comparison and post-publication analysis of large datasets by independent researchers. Furthermore, this approach also facilitates collaboration to increase replicability/reproducibility and re-use of cognitive data and ultimately increases the accuracy of predictions regarding cognitive phenotypes and outcomes in drug efficacy studies.

Currently, several different species can be tested using touchscreens for cognitive assessment, including rats, primates, monkeys, birds, and dogs (*Bussey et al., 2008*; *Charles et al., 2004*; *Guigueno et al., 2015*; *Horner et al., 2013*; *Kangas and Bergman, 2012*; *Mar et al., 2013*; *Nagahara et al., 2010*; *Rodriguez et al., 2011*; *Schmitt, 2018*; *Steurer et al., 2012*; *Wallis, 2017*). Our current scripts can facilitate the formatting of files from such studies and ultimately data from different species, including rats, will be easily incorporated into MouseBytes or similar platforms. Moreover, one can easily envision outputs from unidentified human touchscreen cognitive testing being stored and accessed using similar repository and Web applications. Given the potential for identical touchscreen tests in mice and humans, these data may prove valuable for understanding the consequences of specific mutations for high-level cognition (*Nithianantharajah et al., 2015*).

A major publication bias is the lack of published null datasets, which are important to avoid waste of resources. MouseBytes provides a platform for the dissemination of datasets for touchscreen cognitive assessment even when results show no change in high-level cognition. We anticipate that researchers using automated touchscreen tests will benefit by making their original data available for the community as an integral part of scientific record and publication. This database will become exponentially more valuable when data from more strains of mouse models of disease, drug treatments and genetic manipulations are deposited. Furthermore, as an open source, MouseBytes will be built as a platform where the research community can contribute to new features and share new codes for data analysis. Indeed, MouseBytes is part of a large open initiative for the touchscreen/cognitive behaviour community which includes the touchscreencognition.org platform as well, a knowledge sharing platform that allows storage of protocols and community-driven discussions.

We envision that with MouseBytes it will become easier to connect transcriptomic and different modalities of imaging data from mouse models to their cognitive performance. Ultimately, the integration of current and new touchscreen tests with the use of MouseBytes will change how cognitive function is evaluated in rodents facilitating the discovery of new therapeutic approaches for neurodegenerative and neuropsychiatric disorders.

## Materials and methods

**Key resources table**

| Reagent type (species) or Resource | Designation | Source or reference | Identifiers | Additional Information |
|---|---|---|---|---|
| Antibody | 6E10 Primary Antibody (Human Monoclonal) | Covance | RRID:AB_2564652 Lot#: D13EF01399 Cat#: SIG-39320 | IF (1:200) |
| Antibody | 488 Goat Anti-Mouse Secondary Antibody (Mouse Polyclonal) | Invitrogen | RRID:AB_2534069 Cat#: A-11001 | IF(1:1000) |
| Commercial Assay Kit | Amyloid Beta 42 Human ELISA Kit – Ultrasensitive | Invitrogen | Cat#:KHB3544 | |

*Continued on next page*

*Continued*

| Reagent type (species) or Resource | Designation | Source or reference | Identifiers | Additional Information |
|---|---|---|---|---|
| Strain | B6.Cg-Tg[APPswe, PSEN1dE9] 85Dbo/Mmja (Mouse, Male, Female) | Jackson Laboratories | RRID:MGI:034832-JAX Stock#: 034832-JAX | |
| Strain | (B6;129-Psen1$^{tm1Mpm}$ Tg[APPSwe,tauP301L-1Lfa 0 (Mouse, Male, Female) | Jackson Laboratories | RRID:MGI:101045-JAX Stock#: 101045-JAX | |
| Strain | B6SJL-Tg(APPS wFlLon,PSEN1*M 146L*L286V)6799Vas/Mmja (Mouse, Male, Female) | Jackson Laboratories | RRID:MGI:034840-JAX Stock#: 034840-JAX | |
| Strain | C57BL/6 (Mouse, Male, Female) | Jackson Laboratorie | RRID:MGI:000664-JAX Stock#: 000664 | |
| Strain | B6129SF1/J(Mouse, Male, Female) | Jackson Laboratorie | RRID:MGI:101043 Stock#: 101043 | |
| Strain | B6SJLF1/J(Mouse, Male, Female) | Jackson Laboratorie | RRID:MGI:100012 Stock#: 100012 | |
| Software | ABET II Touch | Lafayette Neuroscience | Model#: 89505 | |
| Software | Spotfire | TIBCO | https://www.tibco.com/products/tibco-spotfir | |
| Software | Touchscreen Quality Control Syste | BrainsCAN | https://github.com/srmemar/Mousebytes-QualityControl | |

## Contact for reagent and resource sharing

All cognitive data are deposited in MouseBytes (www.mousebytes.ca). Further information and requests for resources and reagents should be directed to and will be fulfilled by the Lead Contact, Marco Prado.

## Experimental model and subject details

### Animals

The choice of AD mouse lines considered mice with different rates of accumulating pathology, their use by a variety of researchers, and commercial availability from a single source. Three AD mouse lines were tested: 3xTG-AD (B6;129-Psen1$^{tm1Mpm}$ Tg[APPSwe,tauP301L-1Lfa 0] – RRID:MGI:101045-JAX) and its age-matched control mice (B6129SF2/J RRID:MGI:101045); 5xFAD (B6.Cg Tg [APPSwFl-Lon,PSEN1*M146L*L286V] 6799Vas/J RRID:MGI:034840-JAX) and control mice B6SJLF1/J (RRID: MGI:1000120; and APP/PS1 [(APPswe PS1dE9 B6.Cg-Tg[APPswe,PSEN1dE9]85Dbo/Mmjax RRID: MGI:034832-JAX) and control mice C57Bl/6J (RRID RRID:MGI:000664). All mice used in this study were bred at the Jackson Laboratory in Bar Harbor, Maine and shipped to Canada. Procedures were conducted in accordance with approved animal protocols at the University of Western Ontario (2016/104) and the University of Guelph (3481) following the Canadian Council of Animal Care and National Institutes of Health guidelines. The N values for each group of animals can be found in *Supplementary file 1*. We did not formally calculate power analysis a priori. Typical experiments using the Bussey-Saksida touchscreen technology use samples sizes between 7 and 13 mice per group (*Beraldo et al., 2015*; *Kolisnyk et al., 2015*; *Kolisnyk et al., 2013*; *Lim et al., 2019*; *Romberg et al., 2013b*; *Romberg et al., 2011*). Based on these previous studies, and depending on cohort availability, we assigned a minimum of 10 animals per genotype, sex and site, anticipating some mortality. Due to death of mice, which was a particular problem for APP/PS1, final numbers for some groups were lower than the initial starting numbers (see *Supplementary file 1*).

## Animal housing and food restriction protocols

Mice were housed at two different sites. Half of the mice used in this study were shipped by Jackson Laboratory and housed at the University of Guelph, (Guelph, ON, Canada), while the other half was housed at The University of Western Ontario (London, ON, Canada). For all the 5-CSRTT experiments, an equal distribution of male and female mice was kept at both sites, allowing for comparison and reproducibility studies. For PD and PAL experiments, all male mice were tested at The University of Western Ontario and all the females were tested at the University of Guelph. At each site, mice were housed in a single colony room maintained on a standard 12 hr light cycle (8 am lights on, 8 pm lights off). All the experiments were conducted during the light phase of the cycle. The colony rooms were typically maintained at a temperature of 22–26°C. Mice were initially housed in groups of two mice per cage (16 cm x12cm x 26 cm) and they had their tail tattooed with a unique animal identification number. During each experiment, mice were maintained on a restricted food diet to ensure adequate levels of motivation and to maintain their body weight at 85% of their original weight. Male mice were fed on average 2.5 g of chow, while female mice were given 2.0 g per day (Tekland Chow – Harlan). We set a maximum initial weight of 25 g and if a mouse was over that weight we slowly decreased its weight to 25 g and then set this as a 100% (initial weight). Mice were weighed every other day to ensure maintenance of body weight at 85% of original weight and water was available ad libitum throughout the course of the experiment. After each session, mice received food according to their body weight. Mice are social animals, but it has been described previously that group housed mice can exhibit aggressive behaviour towards their cage mates and establish social hierarchy. This can also affect, in the non-dominant mouse, gene expression and induces depression and anxiety-like behaviours (*Horii et al., 2017*). After few weeks on food restriction, all the mice in the study (for all three strains) had to be separated and singly housed due to fighting.

### 3xTG-AD mice

3xTG-AD mice present three mutations associated with familial AD forms: human familial AD amyloid-beta precursor protein (APP$^{SWE}$), microtubule-associated protein tau (P301L) and presenilin1 (M146V) (PSEN1, APP$^{SWE}$, and tau$^{P301L}$ as previously described) (*Oddo et al., 2003*). Briefly, single-cell embryos, harvest from homozygous presenilin$^{M146V}$ knocking (129/C57BL6 background) mice, were co-microinjected with human mutant tau$^{P301L}$ and the double mutant APPSwe (MK670/671 NL). Both are under the control of Thy1.2 expression cassette (*Berardi et al., 2005*). Age, sex and genetic background-matched controls (B6129SF2/J) were used. The mice were between twelve to sixteen weeks of age at the start of behavioural testing. At this initial age 3xTG-AD mice exhibit few or no extracellular Aβ deposits and relatively low levels of hyperphosphorylated tau (*Berardi et al., 2005*).

### 5xFAD mice

The generation of the 5XFAD mice by Oakley and colleagues in 2006 has been described previously (*Oakley et al., 2006*). Briefly, 5xFAD mice overexpress three familial AD mutations in human APP (695); the K670N/M671L (Swedish - APP$^{SWE}$), I716V (Florida - APP$^{FL}$), and V717I (London - APP$^{LON}$) mutations. In addition, these mice express the M146L and L286V mutations in human PSEN1. Transgene expression is driven by the mouse neuron specific Thy1 promoter (*Oakley et al., 2006*). These five familial, AD mutations are additive in driving Aβ overproduction (*Oakley et al., 2006*). 5xFAD mice present intraneuronal Aβ starting at 1.5 months of age (*Oakley et al., 2006*). 5xFAD mice and wild-type control males and females were twelve weeks of age at the start of behavioural testing. 5xFAD mice have a mixed background: C57Bl6 and Swiss Jim Lambert (SJL). SJL mice are homozygous for the recessive *Pdeb$^{rd1}$* allele, which codes for the β-subunit of cGMP phosphodiesterase on mouse chromosome 5 (*Clapcote et al., 2005*; *Giménez and Montoliu, 2001*). Thus, F1 5xFAD mice should be heterozygous for the mutation. The mutated allele is a nonsense mutation that decreases the transcription of the phosphodiesterase, leading to retinal degeneration and blindness by wean age at approximately 3 weeks and rendering mice homozygous for the *Pdeb$^{rd1}$* allele unsuitable for use in some experiments (*Giménez and Montoliu, 2001*). This same mutation is seen in FVB/NJ (Friend Virus B/National Health Institute Jackson) mice as well (*Giménez and Montoliu, 2001*). To confirm the absence of homozygous mice for the *Pdeb$^{rd1}$* allele we genotyped samples of 5xFAD mice and controls. Briefly, DNA was extracted from mouse ear tissue and amplified using the

REDExtract-N-Amp Tissue PCR Kit Protocol (Sigma-Aldrich, Oakville, Ontario). Polymerase chain reaction (PCR) was done using the Bio-Rad T100 Thermal Cycler (Bio-Rad Laboratories, Hercules, California) with a 500 bp x 40 cycle schedule (94°C x 3 min followed by 40 x [94°C x 30 s] + 60°C x 30 s + 72°C x 30 s then 72°C x 2 min). The tubes were held at 10°C until use. The following reagents were used for each sample: 5 µl of 2x premix, 0.5 µl of retinal degeneration (RD) three oligonucleotide primer (concentration: 0.5 µM; 28-mer, 5'-TGACAATTACTCCTTTTCCCTCAGTCTG-3', accession number L02109, nucleotides 84 to 111), 0.1 µl of RD4 oligonucleotide primer (concentration: 0.02 µM; 28-mer, 5'-GTAAACAGCAAGAGGCTTTATTGGGAAC-3', accession number L02109, nucleotides 644 to 617) and 2.9 µl of RD6 oligonucleotide primer (concentration: 14.5 µM; 28-mer, 5'-TACCCACCCTTCCTAATTTTTCTCAGC-3', accession number L02110, nucleotides 2539 to 2512). RD3 and RD4 amplify a 0.55 kb PCR product from the $Pdeb^{rd1}$ mutant allele, while RD3 and RD6 amplifiy a 0.40 kb PCR product from the WT allele (*Giménez and Montoliu, 2001*). The PCR products are then run on an agarose gel along with a 100 bp ladder (Gene DireX, Frogga Bio, Toronto, Ontario) and imaged with FluorChem Q (Alpha Innotec Corp., San Leandro, California). The positive control for $Pdeb^{rd1}$ was ear tissue from a Friend Virus B NIH Jackson mouse (FVB/NJ; Jax stock #001800), an inbred strain of mouse known to be homozygous for the $Pdeb^{rd1}$ mutation. This mouse was purchased from the Jackson Laboratory (Bar Harbor, Maine). The control for the WT allele of $Pdeb$ for this gel was ear tissue obtained from a B6SJLF1/J mouse.

### APP/PS1 mice
APP/PS1 are double transgenic mice expressing a chimera of mouse/human APP (Mo/HuAPP695swe) and a mutant human presenilin 1 (PS1-dE9). Generation of this mouse line has been previously described (*Jankowsky et al., 2004*). Transgene expression is driven by the mouse prion protein (PRP) promoter, which results in expression relatively restricted to the central nervous system.

## Methods details

### Touchscreen operant platform
All the behavioural tests were conducted in the automated Bussey-Saksida Mouse Touchscreen Systems model 81426 (Campden Instruments Limited, Loughborough, EN) (*Horner et al., 2013*; *Mar et al., 2013*; *Oomen et al., 2013*). Mice were trained to operate the touchscreens by a series of shaping procedures for PD, 5-CSRTT and PAL. The screens in the touchscreen chamber were blocked with barriers during the experiment. For the 5-CSRTT, the screen was divided into five partitions (132 pixels x 132 pixesl) that were 50 pixels above the screen. For the PAL task, the screen was divided into three partitions (228 pixels x 228 pixels) that were 50 pixels above the screen. For the PD task, the screen was divided into two partitions (240 pixels x 240 pixels) that were 50 pixels above the screen. All the schedules were designed and pre-installed and the data were collected using the ABET II Touch software v.2.20.3 as previously described (Lafayette Instruments, Lafayette) (*Horner et al., 2013*; *Mar et al., 2013*; *Oomen et al., 2013*).

### Behavioural Pre-Training
Mice (10–12 weeks of age) experienced several pre-training stages (shaping) prior to probes in each task. For the first four days of pre-training, the mice were habituated to the testing chambers (Habituation schedules). On *Day 1 (Habituation 1)*, the mice were placed in the testing chambers for 10 min with the house lights off, with no stimuli displayed and no reward presented. On *Days 2–4 (Habituation 2)*, the mice were placed into the testing chambers for 20 min (Days 2 and 3) and for 40 min (Day 4). On Days 2–4, the reward tray light was turned on and the reward (strawberry milkshake; Saputo Dairy Products, Canada) was presented and paired with a sound (tone) for 280 ms every 10 s. During this phase, the mice should strengthen the association between the reward tray light and tone with the reward; however, no performance criteria were in place at the habituation phase.

Following habituation, the mice were subjected to the 'Initial Touch' schedule (Phase I), which involves pairing the reward with the presentation of stimuli (random images for PAL and PD and a white square for 5-CSRTT) on the touchscreen. In this phase, a single stimulus appears randomly in one of the windows. After 30 s the stimulus is turned off and the illumination of the reward tray light is paired with a tone and delivery of the reward (7 µl of strawberry milkshake). If the mouse touches

the screen during the time that the image is displayed, a reward is immediately presented with a tone. A new trial starts when the mouse collects the reward, and Initial Touch sessions are repeated daily until the subject completes 30 trials within 60 min.

The next stage of pre-training, 'Must Touch' (Phase II) involves displaying a stimulus randomly in one of the windows, as before. However, in this phase, differently from Phase I, the mouse is required to touch the stimulus on the screen in order to receive the reward paired with a tone. If the mouse touches any window other than the one in which the stimulus is present, it receives no reward. Daily sessions are repeated until the mouse completes 30 trials in 60 min. The next phase of shaping introduces the animals to the initiation procedure, a schedule called 'Must Initiate' (Phase III). At the beginning of each trial, the reward tray is illuminated, and the mouse is required to initiate the stimulus delivery by a nose poke into the reward tray. Successful initiation extinguishes the tray light, and a stimulus is presented in one of the windows on the screen. After touching the stimulus and collecting the reward, the mouse is subjected to a 5 s inter-trial interval (ITI – houselights off, reward tray inactive and no stimulus presented) before the illumination of the reward tray light signals the beginning of the next trial. A criterion of 30 correct trials within 60 min must be met in order for the mouse to proceed to the next phase. The last pre-training phase (Phase IV) is called 'Punish Incorrect' and requires the mouse to both initiate and touch the stimulus but if an incorrect choice is made, it receives a 5 s timeout, during which the lights are turned on and no reward is delivered. The mice continue performing this phase of shaping until they are able to obtain at least 80% of trials correct within 60 min for two consecutive days. Intertrial intervals of 5 s for 5-CSRTT and 20 s for PD and PAL were used in all the phases. All the Standard Operant Procedures (SOPs) for 5-CSRTT (*Horner et al., 2013*; *Mar et al., 2013*; *Oomen et al., 2013*), PD (SOP2) and PAL (SOP3) and other touchscreen tasks can be found in the Touchscreen Cognition (www.touchscreencognition.org).

## Pairwise visual discrimination (PD) training

After reaching the criteria on Punish Incorrect schedules, mice were trained on PD Acquisition sessions. In this task, mice must initiate the trials by poking their head into the reward tray after a light signal is displayed. Immediately after exiting the reward tray, two different images appear on the screen. Mice are required to learn that a rewarded response is determined by the correct visual image (S$^+$ correct image and S$^-$ incorrect image) (S4A). If a correct response is made, reward is delivered, but if an incorrect response is made, a 5 s time-out is initiated with activation of the house light. Following the time-out period, mice are required to complete the same trial repeatedly until a correct response is made. These correction trials do not count towards the overall trial count. To move to the next phase a criterion of 24 correct out of 30 trials (2 days in a row) is required.

## Pairwise visual discrimination (PD) baseline

Following the achievement of acquisition criteria, each mouse was subjected to two baseline sessions according to the same schedule used for the acquisition sessions. However, there is no criterion required to pass this stage.

## Pairwise visual discrimination (PD) reversal

After completion of the baseline sessions, each mouse was subjected to 10 consecutive daily sessions to assess reversal learning and cognitive flexibility. In these sessions, the contingencies between stimuli and reward were reversed such that the original S$^+$ was now an S$^-$ and vice versa. There were no performance criteria for reversal sessions.

## Pairwise visual discrimination (PD) maintenance

Following the completion of the reversal sessions, maintenance sessions were run once weekly until each mouse was old enough to begin the subsequent acquisition sessions with new stimuli. This maintenance procedure is identical to the Punish Incorrect schedule previously described. There were no performance criteria, and each session ended after 30 trials were completed or after 60 min had elapsed.

### Five choice serial reaction time task (5-CSRTT) training

After reaching criterion on the Punish Incorrect stage of pre-training (Phase IV of pre-training), mice were trained on the 5-CSRTT, which requires responses to brief flashes of light pseudo-randomly displayed in one of the five response windows on the touchscreen chamber, as described previously (*Beraldo et al., 2015*; *Kolisnyk et al., 2015*; *Romberg et al., 2013b*; *Romberg et al., 2011*). All mice, at both sites (The University of Western Ontario and University of Guelph), were tested 5–6 days per week for 50 trials or 60 min per day, whichever occurred first. Each trial started with the illumination of the reward tray where the mouse was required to poke its head. After a 5–10 s variable delay, one of the windows was illuminated and the mouse had up to five extra seconds (limited hold) following stimulus presentation to respond on the screen in order to make a correct response. If a mouse touched the screen during the variable delay prior to stimulus presentation, the response was recorded as a 'premature response' and the mouse was punished with a 5 s time-out followed by a 5 s ITI. The touchscreen stimulus duration (i.e., the duration for which the window is lit) was initially set to 4 s. The first response to a window during the stimulus presentation or the limited holding period was recorded and initiated the next phase of the trial. If a correct choice was made the reward was presented and any incorrect response was punished with a 5 s time-out followed by a 5 s ITI (*Mar et al., 2013*). Failure to respond to any window by the end of the limited hold period was recorded as an 'omission' and punished with a 5 s time out, followed by the 5 s ITI before the start of the next trial. The mice continued on the 4 s stimulus duration until performance was stabilised at greater than 80% accuracy, less than 20% omissions, and 30–50 trials completed for three consecutive days. Once mice reached criteria, training continued on the same task, but with a 2 s stimulus duration; criteria for this phase were the same as for the 4 s version. If mice failed to reach criteria within 30 sessions, they were eliminated from the study.

### Five choice serial reaction time task (5-CSRTT) probe sessions

Once mice reached criterion on the 2 s variation of the 5-CSRTT, they were exposed to a series of probe sessions. During each probe session, one of four test stimulus durations was used: 1.5 s, 1.0 s, 0.8 s, and 0.6 s. Each mouse completed two consecutive days of probe trials with each of the stimulus durations. Following each probe session, mice were returned to the 2 s stimulus duration version for two consecutive baseline days before beginning the next probe session (two days with a different stimulus duration). The order of presentation for probe sessions was counterbalanced across all mice. In order to assess attentional performance longitudinally (which to the best of our knowledge has not been performed in AD mice), the same mice were tested on probe sessions at 4, 7, and 10–11 months of age. To ensure maintenance of sufficient baseline performance on the task between probe sessions, mice were given a single day of training with the 2 s stimulus duration each week until the next set of probe sessions. Prior to the start of the 7 month and 10–11 month probe sessions, mice were given 5 days of 2 s 5-CSRTT sessions in order to re-baseline them on the task. The same reduced stimulus durations (1.5 s, 1.0 s, 0.8 s, and 0.6 s) were used for probe sessions at each testing age, and the order of stimulus duration sessions was counterbalanced between mice at each age as well as for each mouse across each of the testing ages.

### Paired associate learning (PAL) training

After reaching the criteria on Punish Incorrect schedules (Phase IV of pre-training), mice were trained on PAL Acquisition sessions. In this task, mice must initiate the trials by poking their head into the reward tray when the light on the feeder turns on. Immediately after exiting the reward tray, two different images appear in two of three positions on the screen. Mice must learn that each specific visual image is associated with only one correct spatial location on the touchscreen, and only one image per trial is presented in the correct location. If a correct response is made a reward is delivered (7 µL of strawberry milkshake), but if an incorrect response is made, a 5 s time-out is implemented with activation of the house light. Following the time-out period, mice are required to complete the same trial repeatedly until a correct response is made. These correction trials were not counted towards the overall trial count. The criterion for this training phase is the completion of 36 trials in 60 min (per day). All the mice from both groups and all the time points were able to reach this criterion in the first session.

## Paired associate learning (PAL) probe sessions

After successfully completing the training phase, mice were tested either on *different* PAL (dPAL) or on *same* PAL (sPAL) tasks. 3xTG-AD and APP-PS1 were tested in dPAL at 4 and 11 months of age. Due to a poor performance in dPAL at 4 months of age, 5xFAD mice (10–11 months of age) were tested in sPAL, which is relatively easier test when compared to dPAL. In both tasks, a mouse initiates the task by touching the reward tray, which triggers the display of both $S^+$ and $S^-$ on the screen. As described above, $S^+$ refers to the stimulus presented in the correct location and $S^-$ refers to a stimulus presented in the incorrect location. In this task, the mice are required to learn to associate a stimulus with its correct location. Similar to the PAL Acquisition phase, if a correct response is made ($S^+$), the reward is delivered. However, touching the $S^-$ stimulus results in a 10 s time-out and illumination of the light in the chamber (10 s). Following the punishment period, mice are required to initiate the same trial repeatedly until a correct response is made; these correction trials did not count towards the overall trial count. The only difference between dPAL and sPAL is the fact that on dPAL $S^+$ and $S^-$ stimuli are different images and on sPAL $S^+$ and $S^-$ are the same image (*Horner et al., 2013*). Mice were tested on 45 sessions for dPAL or sPAL tasks regardless of performance.

## Longitudinal behavioural testing protocol

Mice were tested longitudinally as they aged at different time points. Different cohorts of 3xTG-AD, 5xFAD and APP/PS1 mice were tested longitudinally starting at approximately 4, 7 and 10–11 months of age on 5-CSRTT and PD and at 4 and 10–11 months of age on PAL. The first time points of testing, for each experiment, were completed when the mice were 5–6 months of age. After the completion of the first set of experiments, all mice were kept on a maintenance schedule (2 s stimulus duration for 5-CSRTT and random images for PD and PAL) once a week until the commencement of the second set of probe trials that started at 7 months of age. The images used for the training and maintenance were removed from the probe trial database and were not displayed to the mice during the probes for PAL or PD tasks. This was done to ensure that the mice did not forget the basic aspects of touchscreen task performance and hence would not require retraining prior to the subsequent 7- and 11 month trials. Upon completion of the second set of trials, mice were put back on a weekly maintenance schedule until the commencement of the third set of trials at 11 months of age.

## Image set control experiments

For PD and PAL, different sets of images were used at each testing age. All the images were randomly selected from the ABET II imaging databank and, according to the manufacturer, the images present the same number of pixels. Image sets I, II and III were used for testing mice on PD at 4, 7 and 10–11 months of age, respectively (S4). Image sets IV, V and VI were used to test mice on dPAL or sPAL at 4 and 10–11 months of age (S4). It is well known that rodents prefer some stimuli over others (*Bussey et al., 2008*). To evaluate the potential stimulus biases, we tested different cohorts of wild-type male mice (B6129SF2/J MMRCC stock number 101045), at 4 months of age, with three different images sets used for testing the WT and transgenic mice on PD (Image sets I, II, and III) and PAL (Image sets IV, V, and VI). (S4). PD testing for image biases was done at The University of Western Ontario, and the PAL testing was done at the University of Guelph.

## Tissue preparation

Mice were anaesthetized using ketamine (100 mg/kg) and xylazine (25 mg/kg) in 0.9% sodium chloride solution, and then transcardially perfused with 1x phosphate buffered saline (PBS, pH = 7.4) for 5 min. For each mouse, one harvested hemibrain was post-fixed in 4% paraformeldahyde overnight and subsequently used for immunostaining, while the other harvested hemibrain was stored at −80° C for biochemical analyses.

The hemibrains for immunohistochemistry were cryopreserved using increasing concentrations of sucrose (15%, 20%, 30%), embedded in optimal cutting temperature (OCT) compound and frozen at −80°C. Sagittal sections (10 μm) were cut using the Cryostat (Leica Biosystems), directly mounted and frozen. All slides were immersed in 70% ethanol for 1 min followed by distilled water for another minute before they are stained as described below.

## Thioflavin-S

Slides were stained with filtered 1.25% Thioflavin-S solution in 50% ethanol for 8 min at room temperature. Slides were then washed twice with 80% ethanol, once with 95% ethanol, and then three times with distilled water before mounting.

## Aβ immunoflourescence

Slides were washed twice with Tris Buffered Saline (TBS) 1x and then permeabilised with 1% Triton X-100 (tx) in 1x TBS for 15 min. Non-specific binding was prevented by incubating the slides for one hour in 2% horse serum (HS) and 2% bovine serum albumin (BSA) in TBS 1x with 0.3% Triton-x. Slides were then stained overnight at 4°C with 6E10 primary antibody (RRID:AB_2564652) diluted 1:200 in TBS 1x. Following two TBS 1x washes, slides were incubated at 4°C in 488 goat-anti-mouse secondary antibody (RRID:AB_2564652) diluted 1:1000 in TBS 1x, 1% HS and 1% BSA. Nuclei were stained with To-Pro-3-Iodide (Life Technologies. Gibco, Carlsbad, CA, USA) diluted in PBS1x (1:1000) for 15 min. Slides were then rinsed three times with TBS 1x and mounted.

## Aβ(1-42) ELISA

For Aβ (1-42) quantification we used food-restricted and non-food restricted 3xTG-AD and 5xFAD (males and females) by six months of age. Hippocampus fractionation was performed as described previously (*Ostapchenko et al., 2015*). ELISA was performed using the ultrasensitive kit for human Aβ(1-42) (cat#KHB3544, ThermoFisher Scientific, Mississauga, ON, Canada).

## Imaging

Mounted slices were visualised by confocal microscopy using Leica-TSC SP8 or SP5 (Leica Microsystems, Wentzler, Germany) (20x/0.75 objective, 488 nm laser and 647 nm laser). Images were analysed using ImageJ (National Institute of Health-NIH, Bethesda, Maryland, USA). For each mouse, the cortex and hippocampus (dentate gyrus, CA3, CA1b and CA1a) of 3–4 slices were imaged and quantified in terms of percentage area. The experimenter was blind to genotype during image acquisition and quantification.

## Quantification and statistical analysis

### Touchscreen data analysis, quality control and storage

To ensure the quality of acquired the data when using touchscreens in a high-throughput mode, several actions have been taken before data processing and analysis. All the procedures (schedules, images, food restriction, database and animal identification nomenclatures, etc.) were standardised between the two sites before the start of experimentation and the SOPs can be found in touchscreencognition.org. Monthly conference calls were made, and reports were exchanged between the researchers to assure the maximum standardisation of the procedures across the sites. After automated collection, the data were automatically saved and backed-up onto two different servers at The University of Western Ontario and the University of Guelph. For data protection, the database and back-up were strictly controlled and logged. ABETT II files were converted to XML files (each XML file corresponds to the data from a unique mouse ID session/day of training or testing) and uploaded into mousebytes.ca. The XML files were then automatically checked by automated quality control (QC) algorithm and the codes are available for free download and modification in GitHub (https://github.com/srmemar/Mousebytes-QualityControl) (*Memar et al., 2019*) Files with potential errors, due to human input and/or machine/software failure, were automatically flagged by the QC procedure. The discrepancies, flagged by the QC, were fixed manually when possible and corrupted files were discarded and not used for data analysis. For example, in most of cases the flagged files were due to a software failure at the beginning of the session or wrong animal ID number input. If there was a software failure at the beginning, a new session was started in the same day. This generates two XML files for the same mouse ID in the same day and the incomplete running session file is flagged and not transferred to the database based on the QC rules. If there is a wrong mouse ID input, MouseBytes will flag the XML file related to the wrong ID and the user is able to enter the right ID so the file can be transferred to MouseBytes. The Interquartile ranges (IQRs) method was used to filter out outliers from the temporal features for each cognitive task. So, any feature value

beyond the sum of third quartile (Q3) and 3*IQRs (i.e. Q3 + 3*IQRs) was considered as extreme out-lier and automatically removed from the dataset (*Parrinello et al., 2016*).

The processed data were transferred to the open-access application (*Parrinello et al., 2016*). The complete data set is also available for visualisation and customised analyses on the analytics platform TIBCO Spotfire (*Dunn et al., 2016*; *Pechter et al., 2016*), integrated in MouseMytes. Guidelines to access and visualise the data on Spotfire and in MouseBytes can be found in mousebytes.ca/spotfire and mousebytes.ca/tutorial.

### 5-CSRTT analysis

In all pre-training stages for the 5-CSRTT task (phases I-IV), the number of sessions to reach criterion were analysed to determine any differences in learning of the task. A similar analysis was performed for both 4 s and 2 s stages of 5-CSRTT training. For the probe sessions, several parameters were analysed as an average between each set of two probe sessions with each stimulus duration: *Accuracy* – percentage of correct trials; *Omissions* – percentage of trials on which no response is made; *Correct Response Latency* – reaction time for correct response; *Reward Collection* Latency – reaction time to collect the reward on correct trials; Premature responses – number of responses made prior to the stimulus presentation; Perseverations – number repeated responses at a previously rewarded window before onset of the next trial. Omissions and Premature responses did not count towards Accuracy. Analysis of the probe trial data was conducted using a 3 (age) x 4 (stimulus duration) x 2 (research site) x 2 (genotype) x 2 (sex) split-plot ANOVA. In addition, vigilance in the 5-CSRTT was analysed with a 4 (stimulus duration) x 5 (block) x 2 (genotype) split-plot ANOVA. To characterise genotype-specific effects, we a priori decided to conduct 4 (stimulus duration) x 2 (genotype) split-plot ANOVA analysis between wildtype and transgenic mice for each sex, strain, and age.

### PD analysis

For the pre-training stages for the PD task (Phases I-IV) and acquisition sessions, the number of sessions to reach criterion were analysed to determine any differences in task learning. In addition, several behavioural parameters were analysed for the PD and reversal phases: *Accuracy* – percentage of correct trials; Correction trials – number of trials until a correct choice is made. *Correct Response Latency* – reaction times for correct response; *Reward Collection* Latency – reaction time to collect the reward on correct trials. To analyse PD reversal data, a 10 (session) x 2 (genotype) x 2 (sex) split-plot ANOVA was conducted. We decided a priori to investigate genotype-specific effects within each mouse line, so we additionally conducted 10 (session) x 2 (genotype) split-plot ANOVA within each sex, strain, and age.

### PAL analysis

In the pre-training stages for the PAL task, the number of sessions to reach criterion were analyzed to determine any differences in basic touchscreen acquisition. In addition, several behavioural parameters were analysed for the dPAL/sPAL and retention phases: *Accuracy* – percentage of correct trials; Correction trials – number of trials until a correct choice is made. *Correct Response Latency* – reaction times for correct response; *Reward Collection* Latency – reaction response to collect the reward on correct trials. Data from the 45 sessions (days) of dPAL/sPAL were binned in intervals of 5 sessions or one week of testing (e.g. sessions 1–5 is binned as week 1, session 6–10 is binned as week 2, etc.). Data for the PAL training was analysed with a 9 (bin) x 2 (sex) x 2 (genotype) split-plot ANOVA. All data were analysed separately at 4 months and 11 months of age and for males and females. To characterise genotype effects specifically, we a priori decided to conduct an additional 9 (bin) x 2 (genotype) ANOVA models separating by sex, strain, and age.

### Longitudinal k-means clustering

In order to categorise the overall performance of the mice over the duration of the tasks we employed a k-means clustering approach. Given the longitudinal nature of the behavioural tasks, repeated performances across multiple trials, a three-dimensional version of the analysis is required. To do so, we used the kml3d R package (*Genolini et al., 2016*; *Genolini et al., 2013*). This allows for clustering analysis to be performed across trials. The clusters were renamed post-hoc based on the overall performance of the mice on key metrics in the task, which resulted in a high, mid, and

low performing cluster. In order to determine if differences in k-mean group composition existed between wildtype and transgenic mice, Fisher's exact test was conducted.

## Aβ immunofluorescence

For all immunofluoresecence studies, a total of 8 B6SJLF/1 (4 female, four male) and 8 5xFAD (four female, four male) mice were used. Immunofluorescent analysis was conducted on four slices from each brain sample. Comparison of 6E10 and thioflavin-S expression between genotypes was conducted using two-tailed independent samples t-tests.

## Aβ ELISA

A total of 6 B6129SF2/J (three male, three female), 8 B6SJLF/1 (4 female, four male), 6 3xTG-AD (three male, three female), and 8 5xFAD (four male, four female) mice were used for the ELISA analysis of Aβ. In order to quantify differences between wildtype and transgenic mouse Aβ expression, two-tailed independent samples t-tests were used.

## Data and software availability

The data that support the findings of this study are available from the https://mousebytes.ca data repository. The code used for analysis of these data can be found in https://github.com/srmemar/Mousebytes-QualityControl (*Memar et al., 2019*).

## Additional resources

For access to the raw data from these behavioural experiments, upload new experiments, or visualise current experiments, please visit the (*Memar et al., 2019*) data repository.

## Acknowledgements

We thank Chris Gorgolewsky for helpful discussion, Jue Fan and Sanda Raulic for mouse genotyping support, Suro Lee, Hillary Kim, Meghan Thorne, Jocelyn Shubert, Martine Grenon, and Theresa Martin, for the help with mouse weighing and feeding.

This work was supported by the Weston Brain Institute (Canada), Canadian Institute of Health Research (MOP136930, MOP126000 and MOP89919), NSERC, Alzheimer's Society of Canada, Canadian First Research Excellence Fund (BrainsCAN) and Brain Canada. LMS is supported by CIFAR, SM is a Canadian Open Neuroscience Platform (CONP) Scholar, DP received a Post-Doctoral fellowship from MITACS, LMS and MAMP are Tier 1 Canada Research Chairs.

## Additional information

### Competing interests

Lisa M Saksida: Lisa Saksida has established a series of targeted cognitive tests for animals, administered via touchscreen within a custom environment known as the "Bussey-Saksida touchscreen chamber". Cambridge Enterprise, the technology transfer office of the University of Cambridge, supported commercialization of the Bussey-Saksida chamber, culminating in a license to Campden Instruments. Any financial compensation received from commercialization of the technology is fully invested in further touchscreen development and/or maintenance. Timothy J Bussey: Tim Bussey has established a series of targeted cognitive tests for animals, administered via touchscreen within a custom environment known as the "Bussey-Saksida touchscreen chamber". Cambridge Enterprise, the technology transfer office of the University of Cambridge, supported commercialization of the Bussey-Saksida chamber, culminating in a license to Campden Instruments. Any financial compensation received from commercialization of the technology is fully invested in further touchscreen development and/or maintenance. The other authors declare that no competing interests exist.

## Funding

| Funder | Grant reference number | Author |
|---|---|---|
| Weston Brain Institute | | Robert Bartha<br>Stephen S Strother<br>Boyer D Winters<br>Marco AM Prado |
| Canadian Institutes of Health Research | MOP136930 | Marco AM Prado |
| Alzheimer Society | | Vania F Prado<br>Marco AM Prado |
| Canada First Research Excellence Fund | BrainsCAN | Robert Bartha<br>Lisa M Saksida<br>Timothy J Bussey<br>Vania F Prado<br>Marco AM Prado |
| Brain Canada | Multi-Investigator Research Grant | Vania F Prado<br>Marco AM Prado |
| Canadian Institutes of Health Research | MOP126000 | Vania F Prado<br>Marco AM Prado |
| Canadian Institutes of Health Research | MOP89919 | Vania F Prado<br>Marco AM Prado |
| Natural Sciences and Engineering Research Council of Canada | | Lisa M Saksida<br>Timothy J Bussey<br>Vania F Prado |
| Canada Research Chairs | | Lisa M Saksida<br>Marco AM Prado |
| Brain Canada | Canada Open Neuroscience Platform | Sara Memar<br>Timothy J Bussey<br>Marco AM Prado |
| Mitacs | | Daniel Palmer<br>Lisa M Saksida<br>Timothy J Bussey |
| CIFAR | | Lisa M Saksida |

The funders had no role in study design, data collection and interpretation, or the decision to submit the work for publication.

## Author contributions

Flavio H Beraldo, Conceptualization, Data curation, Formal analysis, Supervision, Investigation, Visualization, Methodology, Project administration; Daniel Palmer, Conceptualization, Data curation, Software, Formal analysis, Investigation, Visualization, Methodology; Sara Memar, Data curation, Software, Visualization, Methodology; David I Wasserman, Conceptualization, Data curation, Investigation; Wai-Jane V Lee, Samantha D Creighton, Chris Fodor, Formal analysis, Investigation; Shuai Liang, Tom Gee, Data curation, Software; Benjamin Kolisnyk, Data curation, Software, Formal analysis; Matthew F Cowan, Justin Mels, Talal S Masood, Mohammed A Al-Onaizi, Investigation; Robert Bartha, Conceptualization, Resources, Funding acquisition; Lisa M Saksida, Vania F Prado, Timothy J Bussey, Stephen S Strother, Boyer D Winters, Marco AM Prado, Conceptualization, Resources, Supervision, Funding acquisition

## Author ORCIDs

Flavio H Beraldo (iD) https://orcid.org/0000-0001-7638-2007
Daniel Palmer (iD) https://orcid.org/0000-0002-3419-8647
Stephen S Strother (iD) https://orcid.org/0000-0002-3198-217X
Boyer D Winters (iD) https://orcid.org/0000-0003-3613-4441
Marco AM Prado (iD) https://orcid.org/0000-0002-3028-5778

### Ethics

Animal experimentation: Procedures were conducted in accordance with approved animal protocols at the University of Western Ontario (2016/104) and the University of Guelph (3481) following the Canadian Council of Animal Care and National Institutes of Health guidelines.

### Decision letter and Author response

Decision letter https://doi.org/10.7554/eLife.49630.sa1
Author response https://doi.org/10.7554/eLife.49630.sa2

## Additional files

### Supplementary files

• Supplementary file 1. The N values for each group of animals used in this study.

• Supplementary file 2. Summary of split-plot ANOVA of all behavioural measures for each genotype for 5-CSRTT.

• Supplementary file 3. Summary of split-plot ANOVA of all behavioural measures for each genotype for PAL.

• Supplementary file 4. Summary of split-plot ANOVA of all behavioural measures for each genotype for PD.

• Supplementary file 5. Summary of ANOVA analyses characterising genotype specific effects for 5-CSRTT.

• Supplementary file 6. Summary of ANOVA analyses characterising genotype specific effects for PAL.

• Supplementary file 7. Summary of ANOVA analyses characterizing genotype specific effects for PD.

• Supplementary file 8. Vigilance analysis characterising mice performance across blocks of 10 trials in 5-CSRTT.

• Transparent reporting form

### Data availability

Automated quality control (QC) algorithm and the codes are available for free download and modification in GitHub https://github.com/srmemar/Mousebytes-An-open-access-high-throughput-pipeline-and-database-for-rodent-touchscreen-based-data (copy archived at https://github.com/elifesciences-publications/Mousebytes-An-open-access-high-throughput-pipeline-and-database-for-rodent-touchscreen-based-data). The touchscreen processed data were deposited into an open-access application (http://www.mousebytes.ca/).

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
