## [Decision Letter]

**Acceptance summary:**

Neuroscience is currently benefiting from rapid technological advances that enable the monitoring and manipulation of neural systems and circuits with unprecedented power. However, progress in designing novel approaches to measure, collate and analyze behavioral correlates of neural function has lagged these developments. Here, the authors introduce an open-access online platform and data repository to facilitate the analysis and sharing of large-scale datasets obtained from an increasingly widely-used computerized, touchscreen-based, apparatus for assaying a range of cognitive processes in rodents. They go on illustrate the utility of this resource for revealing novel sex- and genetic-background related cognitive domains in a multi-site cohort of Alzheimer's disease model mice. This publication serves as a timely example of the potential for open-science in an era in which establishing effective networks connecting investigators and their growing repositories of ('big') data is becoming increasingly important.

**Decision letter after peer review:**

Thank you for submitting your article "An open-access high-throughput pipeline and database for rodent touchscreen-based cognitive assessment" for consideration by *eLife*. Your article has been reviewed by three peer reviewers, and the evaluation has been overseen by a Reviewing Editor and Huda Zoghbi as the Senior Editor. The following individuals involved in review of your submission have agreed to reveal their identity: John G Howland (Reviewer #2); Yogita Chudasama (Reviewer #3).

The reviewers have discussed the reviews with one another and the Reviewing Editor has drafted this decision to help you prepare a revised submission.

Summary and Essential revisions:

All three reviewers expressed enthusiasm for efforts to standardize and share data generated by this increasingly popular behavioral testing paradigm. The major concerns raised at this stage for the authors to address relate to: 1) the ease of 'adoptability' of the analysis across laboratories that are not working in collaboration with the authors, and 2) the strength of the narrative justifying the collation of the analysis methods with the example empirical data in this one paper. In my opinion, these issues should be addressable in a compelling rebuttal to the reviewers and a carefully revised version of the manuscript.

Reviewer #1:

With more and more researchers adopting touchscreen systems for high-level cognitive assessment, there is certainly a need for a well maintained repository to allow for sharing of published datasets, large meta-analysis, and importantly, but not mentioned, the dissemination of null datasets. Co-authors and institutions involved in the current effort are leaders in the field of using this technology and these tools have the potential to lead to advancements in the field. However, the structure of the manuscript, and in particular the lack of mention of major difficulties in comparing datasets across labs, make the current results unclear, and the tools, such as the K-mean classification approach, difficult to assess.

- The authors repeatedly stress the strength of the touchscreen systems is the standardization of outcomes. However, even within the 3 tasks the authors use here, there are an incredibly amount of variables, including stimulus choices, inter-trial intervals, trials and sessions per day, reward type and amount, stimulus durations for 5-CSRTT etc. that can vary across experiments and labs. Much more discussion and description needs to be given to how these factors can be controlled for, or at least noted in order for Mousebytes users to make well-designed comparisons across datasets.

- The current study stresses the high-degree of collaboration and communication across sites including pre-standardization, use of the same systems and software, monthly conference calls and collaboration on QC. This can't possibly be expected to happen with an open access database. How will the system account for data collected on behavioral systems from other providers, or open source built systems? Gurley, J. Exp Analysis of Beh 2019; Butler and Kennerly, Behav. Res. Methods 2018 A discussion of how the system can or cannot deal with files using different category labels, different orders of variables, etc. needs to be added.

- The authors state both in the Introduction and Materials and methods that during QC "files" with potential errors are flagged by the procedure, fixed manually, or discarded and not used for analysis if not fixable. What is a "file"? Is this a daily data file for an individual animal? If so, how does the software account for the "hole" in the data? Are those data points simply ignored, wouldn't this decrease the total errors, alter the percent correct for that animal?

- While the authors correctly note the huge number of graphs and figures that can be generated from this dataset, the data included in Figure 2 seem arbitrary. Site comparison is included for 3-6 M/Males but not 11-13. Strain is shown for males and not females. Given only 4 figures are included in the main document, there is room to add these panels and aid the reader in the logical assessment of the variables.

- Application of unbiased assessment via machine learning is exciting and potentially important, but the approach is not clear. If the authors wish to unbias the approach, why were sex and age specifically conducted separately? Does this analysis require the assumption of independent samples? Inclusion of each animal as a separate data point at each age would violate this.

- The data presented in Figure 3 (or 4?) do not seem to support the authors' conclusions in the text. The statement "5xFAD and 3xTG-AD transgenic mice consistently are lower performers than their WT counterparts in the 5-CSRTT" is confusing, Based on the figure, it seems that WT 3xFAD males and females are consistently worse at most ages. APP/PS1 results are not discussed, although females show almost no differences (except at 7 months) and in males, WT seem to be consistently worse than TG.

- Follow up motor testing on the 5xFAD males is potentially interesting but detracts from the main point of the manuscript. It is unconvincing that increased reward collection latency "uncover(ed) unexpected phenotypes in mouse models" as 1) general phenotyping of mouse models for characteristics (motor, sight, motivation) that would alter touchscreen results should be done prior to testing and 2) this phenotype was previously reported as the authors note.

- Similarly, the inclusion of the lack of effect of AB burden based on ad lib vs. food restriction is potentially interesting, but here just adds more density to an already overpacked manuscript

- The longitudinal analysis is potentially powerful, but also raises questions. Some discussion of how previous training might alter later performance needs to be included. There is a strong case to be made that touchscreen testing could be considered enrichment. How does previous experience alter later cognitive performance in these models versus standard housed non-tested mice?

Reviewer #2:

Beraldo and colleagues present a novel open source analysis pipeline for data coming from a suite of tasks used in touchscreen-equipped operant conditioning chambers. They use this pipeline to analyze the behaviour of 3 mouse lines used to study Alzheimer's disease. The pipeline and on-line interface provide an exciting opportunity for combining research from different laboratories to increase understanding of cognition using touchscreen-equipped operant conditioning chambers. Results from the mice tested in the present experiments show results related to the reliability of touchscreen-based assessments, the effects of genetic background, sex, and age on cognition, and also provide an example of using K-mean clustering to analyze behaviour of hundreds of mice in the 3 tasks. Overall, the study is important, exciting, and will provide a useful resource for behavioural neuroscientists studying cognition in rodents. Below are a series of queries for the attention of the authors.

The title of the paper indicates that the pipeline will be used to study "rodent touchscreen-based cognitive assessment". However, the manuscript deals exclusively with mice. The absence of discussion of rats (or other species) appears to be a missed opportunity in this regard. Will the MouseBytes site support datasets from researchers studying rats, or other species?

The Introduction of the manuscript provides a compelling case for the importance of open science and the researchers should be lauded for pushing forward with this pipeline related to cognitive assessment. However, a more detailed discussion of the hardware used to collect the data should be provided. As the hardware used at both sites was the Bussey-Saksida Mouse Systems, it would of considerable interest for other open source hardware options were discussed (e.g., https://labrigger.com/blog/2018/03/09/custom-touchscreen-behavior-systems-for-rodents/). Will the data coming from these systems be compatible with the MouseBytes pipeline?

There are two important components to this study: 1) the introduction of MouseBytes; and 2) the results of assessing the AD-related mice. I found that the specifics of the findings related to the AD-mice were lost in places, particularly the summary and end of the Introduction. How do the authors' experimental results advance our understanding of the 3 mouse models and what specific future research questions can/should be answered with them?

Reviewer #3:

This paper is essentially a methods paper. The authors have taken a bold step in putting together a much warranted open-access database that enables transparency, collaboration and potential likelihood of reproducibility. The paper is well written with clear detail of the database repository, mousebytes.ca.

I have only two comments which are essentially suggestions that I think might help make the paper stronger.

1) The question is how far can mice carry Alzheimer Disease research or any other disease related research, and is data sharing and transparency the answer to translation? The growing disillusionment with murine models of human disease especially in drug discovery, have questioned mouse models constructed with highly penetrant alleles of human disease because of the large number of compounds which apparently 'cure' the mouse but not the human. This discrepancy is repeatedly argued to be due to failure in data/experimental reproducibility causing journals to emphasize more statistical details as if statistics will provide greater translation. Seems to me that the mousebyte.ca database is one approach that could feasibly enable a better mouse to human translation or at least work in this direction. The truth of the matter is that it is virtually impossible to replicate data even in the same lab, let alone between labs for a multitude of reasons. Can mousebytes.ca facilitate translation?

2) At the moment, the paper focuses on three mouse lines thought to model Alzheimers disease (AD) to highlight the strength of the database. The database is also specific to data acquired using the operant touchscreen platform because non-automated methods, the authors argue, are subject to large variations. Are their plans to expand the database to a much larger cohort of mice that are not necessarily tested for cognitive behaviors tested on a touchscreen platform? The large array of autistic mouse models come to mind for which social communication and interaction, motor, cognitive and emotional behaviors are all variable leaving a very confused state in the field.

---

## [Author Response]

Summary and Essential revisions:All three reviewers expressed enthusiasm for efforts to standardize and share data generated by this increasingly popular behavioral testing paradigm. The major concerns raised at this stage for the authors to address relate to: 1) the ease of 'adoptability' of the analysis across laboratories that are not working in collaboration with the authors, and 2) the strength of the narrative justifying the collation of the analysis methods with the example empirical data in this one paper. In my opinion, these issues should be addressable in a compelling rebuttal to the reviewers and a carefully revised version of the manuscript.

We thank the editors for summarizing the main concerns and the reviewers for the positive evaluation. We have addressed these two concerns below in the response to reviewers.

Regarding point 1, the intrinsic features of MouseBytes, and particularly those now added in response to the reviewers’ comments, are specifically designed to promote ease of use for all touchscreen users, especially those with whom we do not collaborate directly. MouseBytes allows users to enter Metadata that facilitates comparison and combination of data from different sites and also different touchscreen apparatuses (see reply to reviewers’ comments below). It is also important to emphasize that MouseBytes is by no means immutable; it will continue to evolve based on user’s feedback.

Thanks to the feedback provided by the reviewers, we made substantial changes to the manuscript, and also to the software. We now can collect even more Metadata on, e.g., experimental conditions. We have also developed codes to easily convert output files from any touchscreen apparatus to the MouseBytes format.

Regarding point 2, we improved the narrative concerning the empirical data, and in particular made it clearer how the data illustrate the advantages of MouseBytes. This is now explained more clearly in the Abstract and Introduction.

Reviewer #1:With more and more researchers adopting touchscreen systems for high-level cognitive assessment, there is certainly a need for a well maintained repository to allow for sharing of published datasets, large meta-analysis, and importantly, but not mentioned, the dissemination of null datasets. Co-authors and institutions involved in the current effort are leaders in the field of using this technology and these tools have the potential to lead to advancements in the field. However, the structure of the manuscript, and in particular the lack of mention of major difficulties in comparing datasets across labs, make the current results unclear, and the tools, such as the K-mean classification approach, difficult to assess.

We thank the reviewer for raising these important points. We have now mentioned throughout the manuscript the potential difficulties in comparing datasets. In particular, we now emphasize collection of Metadata by MouseBytes as a way to greatly facilitate comparison of datasets. Users depositing data can now specify in detail all conditions of the experiments, and further are able to link datasets to published manuscripts which yield further details. Examples include added fields in the repository to account for potential variations in conditions between different laboratories such as light-dark cycle, and group v single housing. The greater availability of these details will increase the power of meta-analysis using MouseBytes.

We have also improved the discussion and explanation about the k-mean analysis. We clearly state now in the manuscript both in the Introduction and Discussion the utility for depositing null-datasets.

- The authors repeatedly stress the strength of the touchscreen systems is the standardization of outcomes. However, even within the 3 tasks the authors use here, there are an incredibly amount of variables, including stimulus choices, inter-trial intervals, trials and sessions per day, reward type and amount, stimulus durations for 5-CSRTT etc. that can vary across experiments and labs. Much more discussion and description needs to be given to how these factors can be controlled for, or at least noted in order for Mousebytes users to make well-designed comparisons across datasets.

The reviewer is right regarding the potential for many variables in any task to be changed using touchscreens. At the moment, major variations in these parameters are identified by the QC codes, and therefore experiments that deviate from standard operating parameters are flagged and can only be uploaded after communication between the user and administrator (subsections “Open-Access Database and repository” and “Touchscreen operant platform” describe these procedures). When experiments deviate from standard parameters, users are requested to provide information in the experiment field and also link the datasets to the published manuscript or preprint providing further details regarding the experimental conditions. Datasets that differ from standard protocols will still be uploadable in MouseBytes and can then be identified and curated for meta-analysis by using the information provided by the experimenter and the QC codes can be also modified by users. This is mentioned in the Results and Discussion and Materials and methods (all new information labelled).

- The current study stresses the high-degree of collaboration and communication across sites including pre-standardization, use of the same systems and software, monthly conference calls and collaboration on QC. This can't possibly be expected to happen with an open access database. How will the system account for data collected on behavioral systems from other providers, or open source built systems? Gurley, J. Exp Analysis of Beh 2019; Butler and Kennerly, Behav. Res. Methods 2018 A discussion of how the system can or can not deal with files using different category labels, different orders of variables, etc. needs to be added.

We would like to thank the reviewer for this suggestion. It has always been our intention to allow Mousebytes to incorporate data not only from Bussey-Saksida touchscreens systems, but from any variant of touchscreen operant chambers as long the output data can be made to match the Mousebytes specifications. We have now developed codes to do just that, and have added this MouseBytes feature to the manuscript. These codes can be downloaded and modified by different users to convert XML files from any touchscreen system output to files that can be readable by MouseBytes (https://github.com/srmemar/Mousebytes-An-open-access-high-throughput-pipeline-and-database-for-rodent-touchscreen-based-data/blob/master/XML_Output.ipynb). This is now described in the Results and Discussion.

- The authors state both in the Introduction and Materials and methods that during QC "files" with potential errors are flagged by the procedure, fixed manually, or discarded and not used for analysis if not fixable. What is a "file"? Is this a daily data file for an individual animal? If so, how does the software account for the "hole" in the data? Are those data points simply ignored, wouldn't this decrease the total errors, alter the percent correct for that animal?

Files are XML outputs generated from a unique session (day) from a unique mouse ID. We have now addressed this point in the subsections “Open-Access Database and repository” and “Touchscreen operant platform”. If a file is flagged and cannot be fixed for some reason, these data are not transferred to MouseBytes and will not be part of the downloadable dataset or used for visualization. In the specific datasets already in MouseBytes, it is unlikely that the discarded files would affect the final results reported in the manuscript. For example, for the 5-choice data less than 0.6% of files were terminally flagged and not used to compose the datasets, and only 0.09% were from probe trials (26 probe trial files flagged files out of 27,440 total files from 5 choice). Hence, the system will not use these data, but it should not have affected the final analysis in the manuscript.

- While the authors correctly note the huge number of graphs and figures that can be generated from this dataset, the data included in Figure 2 seem arbitrary. Site comparison is included for 3-6 M/Males but not 11-13. Strain is shown for males and not females. Given only 4 figures are included in the main document, there is room to add these panels and aid the reader in the logical assessment of the variables.

We thank the reviewer for this suggestion. We now provide several new panels comparing the performance of mice longitudinally, between sites, sex and strains. (Figures 2 and 4) to the current version of the manuscript which is now described in the Results and Discussion

- Application of unbiased assessment via machine learning is exciting and potentially important, but the approach is not clear. If the authors wish to unbias the approach, why were sex and age specifically conducted separately? Does this analysis require the assumption of independent samples? Inclusion of each animal as a separate data point at each age would violate this.

In the k-mean clustering approach that we present in the manuscript, we did not separate sex in the primary clustering algorithm to define Low, Mid and High performers. However, following the k-mean analysis, we determined if group memberships were significantly different by utilizing Fisher’s test. In this secondary analysis, we did separate the data by age and sex. We did this to prioritize looking at group membership and genotype in the Fisher’s Exact Test. In order to address the multiple tests created by separating age and sex, we used a False Discovery Rate correction analyses.

In the k-mean analysis, we assumed that performance could change with age, so age was treated as an independent between factor, where each animal had two or three cluster points for each of the ages of assessment. The k-mean clustering algorithm does not require independence for the initial clustering. The Fisher’s exact test we conducted had independent data points as part of our statistical analysis.

The decision to treat animals at different ages as separate data points was done in order to allow for mice to switch k-mean categories, based on the assumption that performance would change with age. In a given experiment, wildtype mice may naturally have reductions in performance on cognitive tasks related to age. We were interested in detecting if transgenic mice would shift from a category of “high performers” into a category of “low performers” or vice-versa as they were assessed at later time points. More specifically, we were interested in if this shift from high to low performers was larger for transgenic mice than wildtype mice.

Future efforts with the MouseBytes data system will certainly take advantage of other unbiased classification systems (e.g. Mean Shift, DBSCAN, etc.) with larger datasets to begin to better determine how to classify behavioural data from mouse models.

- The data presented in Figure 3 (or 4?) do not seem to support the authors' conclusions in the text. The statement "5xFAD and 3xTG-AD transgenic mice consistently are lower performers than their WT counterparts in the 5-CSRTT" is confusing, Based on the figure, it seems that WT 3xFAD males and females are consistently worse at most ages. APP/PS1 results are not discussed, although females show almost no differences (except at 7 months) and in males, WT seem to be consistently worse than TG.

These data are presented now in Figure 5. Analysis of the dataset shows that indeed, the 3xTG AD and 5xFAD AD mice are shifted to the lower performance group in the 5-Choice Task. Otherwise, their WT counterpart mice are shifted to the higher and moderate performers. For the APP/PS1 mice we did not observed any difference in the performance of females (comparing WT with the TG) in 5 choice. However, interestingly, APP/PS1 male mice shifted their performance to the high and moderate performers while WT shifted to lower performers in the 5-Choice Task. The presentation and discussion of this information are in the Results and Discussion.

- Follow up motor testing on the 5xFAD males is potentially interesting but detracts from the main point of the manuscript. It is unconvincing that increased reward collection latency "uncover(ed) unexpected phenotypes in mouse models" as 1) general phenotyping of mouse models for characteristics (motor, sight, motivation) that would alter touchscreen results should be done prior to testing and 2) this phenotype was previously reported as the authors note.

We agree with the reviewer and accordingly removed the dataset from the manuscript.

- Similarly, the inclusion of the lack of effect of AB burden based on ad lib vs. food restriction is potentially interesting, but here just adds more density to an already overpacked manuscript

We thank the reviewer for the suggestion. However, given that environmental enrichment and decreased food consumption could affect pathology in mouse models, we find these results important for readers to understand that lack of change in cognitive domains is not related to changes in pathology. We have decreased as much as possible the discussion on these issues and only provide information for readers to get the main message.

- The longitudinal analysis is potentially powerful, but also raises questions. Some discussion of how previous training might alter later performance needs to be included. There is a strong case to be made that touchscreen testing could be considered enrichment. How does previous experience alter later cognitive performance in these models versus standard housed non-tested mice?

This is an important point raised by the reviewer. We elaborated on this point in the manuscript (Results and Discussion), but the present work did not directly address this question.

However, one possible source of insight into this question might be in the data from mice performing the 5-Choice serial reaction time task. In our manuscript, we present data from the same mice tested at 3 ages, with final timepoint at 11-13 months, when mice would have been probed twice in this task. Romberg and colleagues, 2011, also tested a 3xTG-AD mice previously with the 5-Choice task at 10 months of age, however, these mice were naïve when tested at this age. The data and results between these two experiments are very similar, suggesting that prior cognitive training may not have a dramatic effect on subsequent touchscreen behavior in the 5-Choice test. Moreover, comparison of performance of mice longitudinally in the 5-Choice task suggests that accuracy does not change with age, although omission is improved after repeated training (See datasets from Figure 2 and Figure 4 and compare 2-6- and 11-13-month-old mice).

We have also analyzed the performance in pairwise visual discrimination and reversal with distinct images in different ages and found that depending on the image mouse performance varied, but previous training did not seem to affect the performance. This is now discussed in the Results and Discussion. We anticipate that with increased datasets being deposited, this important question will be addressed in the future for multiple tasks.

Reviewer #2:[…] Below are a series of queries for the attention of the authors.The title of the paper indicates that the pipeline will be used to study "rodent touchscreen-based cognitive assessment". However, the manuscript deals exclusively with mice. The absence of discussion of rats (or other species) appears to be a missed opportunity in this regard. Will the MouseBytes site support datasets from researchers studying rats, or other species?The Introduction of the manuscript provides a compelling case for the importance of open science and the researchers should be lauded for pushing forward with this pipeline related to cognitive assessment. However, a more detailed discussion of the hardware used to collect the data should be provided. As the hardware used at both sites was the Bussey-Saksida Mouse Systems, it would of considerable interest for other open source hardware options were discussed (e.g., https://labrigger.com/blog/2018/03/09/custom-touchscreen-behavior-systems-for-rodents/). Will the data coming from these systems be compatible with the MouseBytes pipeline?

As described in response to reviewer 1, we have created a way for Mousebytes to incorporate data not only from Bussey-Saksida touchscreens systems, but from any variant of touchscreen operant chambers. To do that, we have developed codes that can be downloaded and modified by different users to convert XML files from any touchscreen system output to files that can be readable by MouseBytes (https://github.com/srmemar/Mousebytes-An-open-access-high-throughput-pipeline-and-database-for-rodent-touchscreen-based-data/blob/master/XML_Output.ipynb). These codes can be easily adapted by users with data obtained in other touchscreen systems. This is now mentioned in the Results and Discussion.

There are two important components to this study: 1) the Introduction of MouseBytes; and 2) the results of assessing the AD-related mice. I found that the specifics of the findings related to the AD-mice were lost in places, particularly the summary and end of the Introduction. How do the authors' experimental results advance our understanding of the 3 mouse models and what specific future research questions can/should be answered with them?

These issues have now been addressed in the Introduction and the Discussion. We identify common age-dependent attentional deficits in 3xTG-AD and 5xFAD mice. We also found that behavioral flexibility did not seem to be consistently affected in most mouse lines tested. This dataset provides the foundation to test new drugs to correct attentional deficits, which were consistently reproduced in 2 out of 3 mouse lines.

Reviewer #3:[…] I have only two comments, which are essentially suggestions that I think might help make the paper stronger.1) The question is how far can mice carry Alzheimer Disease research or any other disease related research, and is data sharing and transparency the answer to translation? The growing disillusionment with murine models of human disease especially in drug discovery, have questioned mouse models constructed with highly penetrant alleles of human disease because of the large number of compounds which apparently 'cure' the mouse but not the human. This discrepancy is repeatedly argued to be due to failure in data/experimental reproducibility causing journals to emphasize more statistical details as if statistics will provide greater translation. Seems to me that the mousebyte.ca database is one approach that could feasibly enable a better mouse to human translation or at least work in this direction. The truth of the matter is that it is virtually impossible to replicate data even in the same lab, let alone between labs for a multitude of reasons. Can mousebytes.ca facilitate translation?

The reviewer summarizes one of our motivations to create a repository for touchscreen datasets. Databases such as MouseBytes can provide more transparency, to allow others to analyze datasets and confirm experiments. We hope that our database can serve as an interface for researchers interested in studying replication and reproducibility.

Additionally, we are hoping to provide a platform for researchers to identify critical features in the lab environment that may contribute to behavioural variation. We have recently begun to address this by incorporation of new meta-data into the MouseBytes experiment records for factors such as colony light cycle, visual stimuli selection, and type of animal housing. We plan on expanding our meta-data system to include several more factors, but these can also be informed in manuscripts that are linked to datasets via their DOI. Eventually, with enough labs and experiments in the system, we can begin to systematically identify which factors may be contributing to experimental variation, and address the issues of translation and reproducibility.

2) At the moment, the paper focuses on three mouse lines thought to model Alzheimers disease (AD) to highlight the strength of the database. The database is also specific to data acquired using the operant touchscreen platform because non-automated methods, the authors argue, are subject to large variations. Are their plans to expand the database to a much larger cohort of mice that are not necessarily tested for cognitive behaviors tested on a touchscreen platform? The large array of autistic mouse models come to mind for which social communication and interaction, motor, cognitive and emotional behaviors are all variable leaving a very confused state in the field.

: We appreciate the desire and need to expand open platforms of behavioural sharing to non-touchscreen behaviour. At this time, we are not considering expanding our system to non-touchscreen behavior, because most of these conventional methods are not automated and have no standardized outputs. We have chosen to focus on touchscreen behaviour with the MouseBytes data, due to the inherent benefits of these systems for replication, experimental control, and translational validity. We are however interested in expanding the touchscreen repertoire to include tasks that are more emotional/social in nature and could probe different types of deficits in other mouse models. Indeed, we are aware of other researchers who are using videos in touchscreens to assess murine social behaviour. We also anticipate the contribution of datasets by the community of researchers using touchscreens using mouse models of developmental disorders.

Our view would be that although touchscreens are ideally suited for databasing applications, the hope is that other behaviours will eventually join the open science/datasharing movement. We think that MouseBytes could help to lead the way.